# Autoencoder neural networks enable low dimensional structure analyses of microbial growth dynamics

Yasa Baig[1,2], Helena R. Ma [3,4], Helen Xu [2] & Lingchong You [3,4,5] ✉

The ability to effectively represent microbiome dynamics is a crucial challenge in their quantitative analysis and engineering. By using autoencoder neural networks, we show that microbial growth dynamics can be compressed into low-dimensional representations and reconstructed with high fidelity. These low-dimensional embeddings are just as effective, if not better, than raw data for tasks such as identifying bacterial strains, predicting traits like antibiotic resistance, and predicting community dynamics. Additionally, we demonstrate that essential dynamical information of these systems can be captured using far fewer variables than traditional mechanistic models. Our work suggests that machine learning can enable the creation of concise representations of high-dimensional microbiome dynamics to facilitate data analysis and gain new biological insights.

Microbial populations and communities exhibit rich temporal dynamics driven by both species-species and species-environment interactions[1–4]. These community dynamics are critical for community self-maintenance and achieving ecological functions. For example, soil microbial communities surrounding plant roots can overhaul their composition in response to changing symbiotic interactions with plant hosts[1]. In human health, microbial predator–prey interactions can accelerate the acquisition of community-scale antibiotic resistance; also, growth rates of various human gut microbial species are modulated in response to diseases like Type II diabetes and IBS[3,4].

Synthetic biology also relies on the predictable control of the temporal dynamics of engineered microbial populations or communities[5]. For instance, Liao et al. developed a three-member microbial system where negative interactions between the three populations can be used to enhance the genetic stability of the engineered circuits[6]. We too have reported the usage of microbial population dynamics for engineering applications, such as coupling programmed bacterial death with environmental sensing to achieve self-regulation of metabolite production or exploiting spatial partitioning to control community biodiversity[7–9].

A common theme of these studies is the need to predict and control temporal dynamics of microbial communities. This task depends on an effective representation of the temporal dynamics by a properly formulated model. Typically, such analysis has relied on using the formulation of mechanistic models based on prior knowledge. For instance, a logistic model and the Monod model are often used to describe growth of a single population; a generalized Lotka-Volterra (gLV) model is often used to describe multi-species community dynamics[8,10,11]. When properly constrained by experimental data, these models can allow prediction of future growth patterns or inferring biological insights[12–15]. Recently, these models have increasingly been paired with black-box machine learning (ML) techniques, which trade mechanistic interpretability to model more complex biological relationships learned directly from data[16]. For instance, we previously utilized growth curves as training data to predict complex phenotypes such as antibiotic resistance directly from growth dynamics[17]. Other studies strike a compromise, using growth curves to derive preselected dynamical features such as growth rate, steady-state cell density, and lag time which are then fed to machine learning models[18].

Each approach, however, presents limitations in how this information is extracted. Training ML models or fitting mechanistic models using whole growth curves with no preprocessing or dimensionality reduction can lead to spurious fits: fluctuations in data can result from biological regulation or from experimental variability. Ad hoc

[1]Department of Physics, Duke University, Durham, NC, USA. [2]Department of Computer Science, Duke University, Durham, NC, USA. [3]Department of Biomedical Engineering, Duke University, Durham, NC, USA. [4]Center for Quantitative Biodesign, Duke University, Durham, NC, USA. [5]Department of Molecular Genetics and Microbiology, Duke University School of Medicine, Durham, NC, USA. ✉e-mail: you@duke.edu

methods, such as moving-average-based smoothing, have been used to minimize the impact of experimental variability, while maintaining critical biological information[13,17,19]. Wavelet analysis has been used to clean microbial growth data in a more principled manner; it also requires an ad hoc choice of frequency components to preserve[19].

In contrast, techniques where growth curves are reduced to a small set of features are simpler to manipulate, interpret, and use for model training, but potentially suffer from greater loss of biological information than smoothing techniques. The choice of features also reflects a bias of the experimenter on the most important components of growth curves needed for downstream machine learning, which may not have any association with phenotypes of interest. For example, prediction of antibiotic resistance phenotype from common features like growth rate and growth integral alone is less effective than using entire growth curves[17].

Certain ML tools can overcome these limitations. Autoencoder neural networks enable the compression of high dimensional, noisy data sets down to low dimensional representations (or embeddings). These embeddings, while lacking biological interpretability, serve as a low-dimension representation of the original data. These embeddings are optimized by the network during training to retain as much signal from the original data as possible; thus, they enable simplifying data representation without a priori bias. The dimensionality of the embedding can be varied during training to enable retaining more and less information of the original data.

In this work, we demonstrate the use of autoencoders for analysis of microbial community dynamics. Specifically, we use autoencoders to compress simulated and experimental growth curves into low-dimensional embeddings. We show that these embeddings can be used to reconstruct the growth curve with high fidelity. Despite drastic dimension reduction, these embeddings contain sufficient information to differentiate between complex phenotypes such as strain identity and antibiotic resistance, predict microbial dynamics from initial conditions and experimental system variables, and can be mapped directly to interpretable mechanistic growth parameters. They can even outperform whole growth curves on these tasks, suggesting a successful elimination of extraneous noise from the original growth curve without loss of biological information. Moreover, we show that the autoencoders can enable the compression of the community dynamics to fewer variables than needed to parameterize a typical mechanistic model.

## Results

### Autoencoder compression of simulated growth curves

An autoencoder consists of two component networks: an encoder that compresses an initial data vector of dimension $D$ to a latent embedding vector of dimension $E$ ($< D$), and a decoder that maps the embedding back to the data vector (Fig. 1a, Table 1). During training, the network improves agreement between the initial vector $\mathbf{x}$ and the reconstructed vector $\mathbf{x}'$ by updating its internal weights to minimize the mean square reconstruction error between all $N$ samples $\sum_i^N ||\mathbf{x_i} - \mathbf{x_i'}||_2^2$. By imposing $E < D$, the network optimizes the lower-dimensional embedding to encode the most critical information about $\mathbf{x}$ necessary for reliable reconstruction (to pre-specified fidelity), while eliminating spurious features in the input data. After training, each embedding vector serves as a compressed representation of the corresponding input data series. Here, we use an asymmetric autoencoder architecture designed to analyze time series (Methods, Supplementary Fig. 1).

As an illustration, we first applied this method to simulated, single-population growth curves, using the logistic model:

$$\frac{dp}{dt} = \mu(1-p)p \tag{1}$$

where $p$ is the relative abundance of the species and $\mu$ is its specific growth rate.

Using random $\mu$ values, we generated 1000 growth curves as the training data. Each curve consists of 100 points (i.e., $D = 100$). We used these data to train an autoencoder model with $E = 2$ (to facilitate visualization of the embedding). The embeddings fall along a single line (approximately $y = x$) (Fig. 1b), indicating that the autoencoder model "learned" the one-dimensional structure of the raw data.

Next, we considered dynamics of microbial communities simulated using a variant of the gLV model[8]. We generated 9500 growth curves by simulating five-member communities using random parameter values. Each growth course consisted of 100 data points ($D = 100$). For this system, embeddings with $E = 2$ are distributed in multiple directions in the latent space, instead of a single line (Fig. 1c). This increase in latent space dimension usage is expected for the increasing complexity and dimensionality of the underlying system. Additionally, the quality of the final reconstructions drops. This is unsurprising given the high amount information bottlenecking required to embed an underlying high dimensional system into a small latent dimension. Indeed, increasing the embedding dimension to $E = 10$ improves the reconstruction quality (15% vs 5% mean absolute error) (Fig. 1d). These results demonstrate even for higher dimensional, complex microbial communities, the autoencoder could develop low-dimensional representations of the growth dynamics.

### Experimental growth curve compression and reconstruction

We next applied the analysis to experimental growth curves. We consider four separate groups of *Enterobacteriaceae* growth curves (Supplementary Table 1). Each time series consists of one growth curve or growth curves of the strain in multiple conditions concatenated together. For each growth curve, we computed their derivative instead of using the growth curve directly. Using the derivative allows the autoencoder to focus on learning representations for the transient dynamics and deprioritizes preserving information about the steady state of each population (Fig. 2a). Since transient dynamics are more information rich and more variable across different bacterial strains, they better capture phenotypical information[17]. The initial dimension ($D$) of each growth curve varied between 98 and 432 (Supplementary Table 1).

For each group, we repeated our encoding and decoding procedure for increasing $E$ while fixing the hyperparameters of the autoencoder. Like the simulated curves, the experimental growth curves could be compressed to and reconstructed from low-dimensional representations with high fidelity (Fig. 2b). Due to the increasing complexity and variability of experimental curves, the autoencoder used the latent space more fully, as indicated by the increased number of non-zero-variance principal components.

Even at low dimensions, the autoencoder reconstructed the principal topological features of each curve such as the positions of peaks and valleys. With increasing $E$, the reconstructions better retained higher-order features such as peak heights and curvature of the raw data (Fig. 2c). Overall, the total square error $\sum_i^N ||\mathbf{x_i} - \mathbf{x_i'}||_2^2$ between the raw and reconstructed curves decreased as $E$ increased from 2 to 30 (Fig. 2d).

### Strain identity classification

Our analysis shows that, despite their variability, the growth curves share a core temporal structure that can be captured by the autoencoder using much lower dimensional representations. We wondered if the reduced embeddings maintained sufficient information to distinguish between the strain identities of different bacteria, as we have demonstrated in a previous study using full time courses[17,19].

For each of the four growth curve groups (Supplementary Table 1), we generated embeddings corresponding to $E = 2, 3, 5, 10, 20, 25$, and 30. We partitioned each dataset into training and testing sets in a 3:1 ratio. We used the training set to train a binary support vector machine

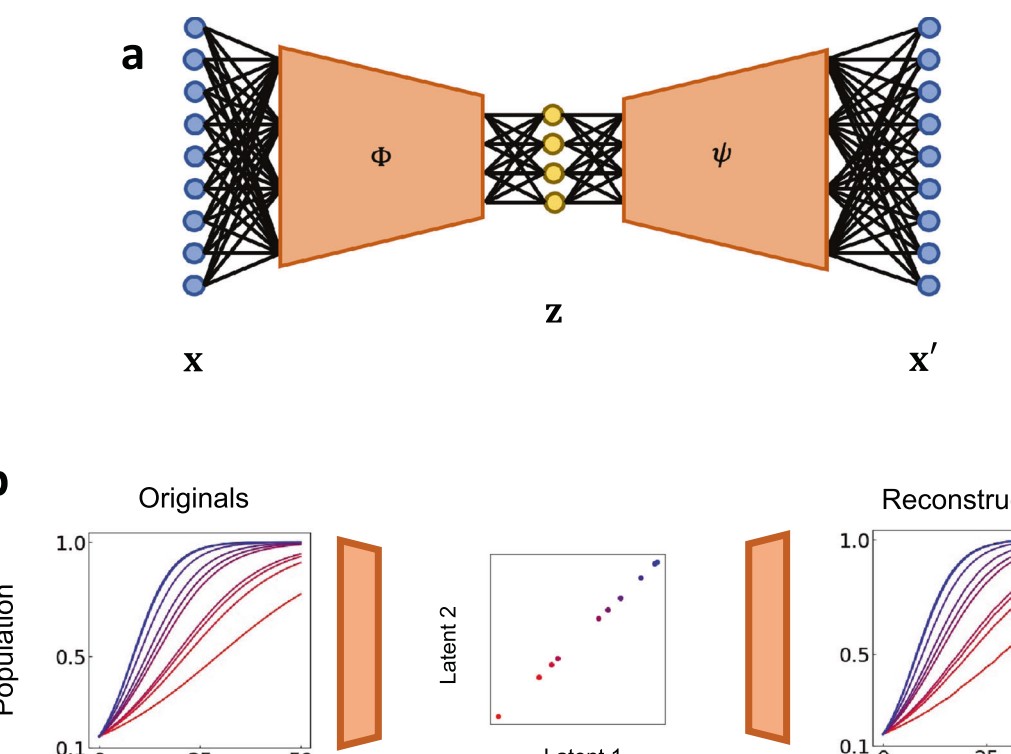

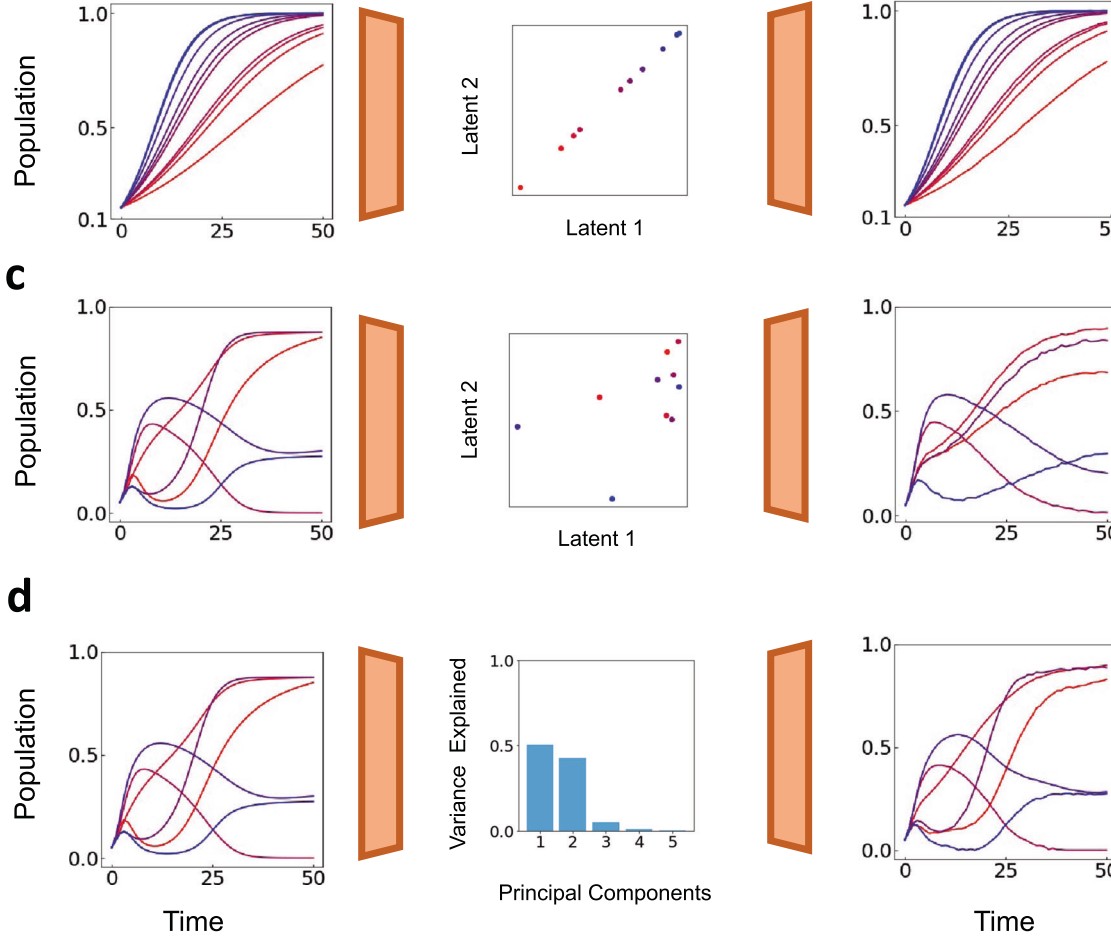

**Fig. 1 | Autoencoder compression of simulated growth curves. a** Autoencoder architecture. An autoencoder consists of two separate neural networks. First the encoder network, $\phi(x)$, maps a time series vector $x$ to a low dimensional, compressed representation in the form of the embedding vector $z$. The decoder network, $\psi(z)$, then attempts to reconstruct $x$ from only the information in $z$. The encoder maps the initial vector through a series of dimension reducing transformations until it reaches the latent layer; the decoder then maps the latent dimension through a series of dimension boosting transformations. This is represented by the size of the encoding narrowing with each layer and the decoder widening with each layer. **b** Embedding of the growth dynamics of a single population, using

$E = 2$. 10 of these curves are shown on the left. The autoencoder embeds all the curves along a single line. The reconstructions of the initial curves are shown on the right. **c** Embedding of the growth dynamics from a five-member community, using $E = 2$. Six curves are shown on the left and compression of these five curves to two dimensions is shown in the center. The reconstructions of the initial curves are shown on the right. **d** Embedding of the growth dynamics from a five-member community, using $E = 10$. The center shows amount of variance explained by each of the first five principal components of the latent space. The original curves are on the left; the corresponding reconstructions are on the right.

**Table 1 | Summary of key terminology and notations**

| Term | Symbol | Definition | Example |
|---|---|---|---|
| Input Dimension | $D$ | The number of time points sampled in a bacterial growth curve used as input to an autoencoder, experimental or simulated. | For experimental growth curve in group 1, the number of features $D = 98$ (Supplementary Table 1). For community simulations, $D$ ranges between 20 and 40. |
| Embedding Dimension | $E$ | The dimension of the latent space to which individual population growth curves or a set of population growth curves corresponding to an entire communities' dynamics are compressed in an autoencoder. | For Fig. 1a, $E = 2$. For Fig. 2b, $E = 30$. For data in Figs. 3 and 5, $E$ is varied as a control parameter. For Fig. 3, $E$ varies between 2 and 30. |
| Community or Phase Space Dimension | $N$ | The number of individual species in a microbial community. Since each species corresponds to an independent variable in the community growth dynamics ODE model, $N$ is also equal to the dimension of the ODE system phase space. | For Fig. 4, $N = 2$. For Fig. 6, $N$ varies between 3 and 6. |
| Parametric Space Dimension | $O(N^2)$ | The number of parameters used to parameterize a growth dynamics ODE model. This varies between choice of underlying ODE model, but generally scales $O(N^2)$. | For Fig. 6, the dimensionality of the parametric space is exactly $N^2 + 1$, with $N^2 - N$ non-zero pairwise interaction terms $\gamma_{ij}$, $N$ growth rates $\mu_i$, and 1 fixed background stress term $\sigma$. |

(SVM) classifier to differentiate between strains (Methods). To achieve multiclass classification, we used a one-vs-all approach where for each strain one SVM classifier was trained to discriminate that strain from all others. Thus, *n* SVMs were trained for a dataset with *n* strains (Fig. 3a). To classify a curve in the test set, we fed it into each trained SVM and selected the strain label that corresponded to the SVM with the highest confidence prediction (Methods).

In both training and testing sets and across all four groups, the classification accuracy increased with embedding dimension (Fig. 3b, c, Supplementary Fig. 2). Furthermore, the embeddings matched or surpassed the classification accuracy of using the raw data with $E$ as low as 10. When considering datasets such as Group 2 where each curve contains >300 data points ($D > 300$), using $E = 5$ yielded a testing accuracy of >95% (Supplementary Fig. 2).

### Low-dimensional embeddings outperform raw data in predicting antibiotic resistance

We further tested if these low dimensional compressions could be used to predict phenotypic traits, such as antibiotic resistance. We considered a library of 244 bacterial clinical isolates with known resistance (or lack thereof) to four antibiotics: sulbactam (SAM), trimethoprim–sulfamethoxazole (SXT), gentamicin (GM), and ciprofloxacin (CIP). This library was generated using same clinical samples used in Group 1 ($D = 98$), except now the labels for prediction corresponded to a binary indicator (1 or 0) whether an isolate was resistant to a particular antibiotic.

We generated several embedding datasets from our full-time courses corresponding to embedding dimensions of $E = 2, 3, 5, 10, 20, 25$, and 30 which were then partitioned 3:1 in training and testing sets. The training sets were used to train four separate SVMs, each distinguishing between resistant vs non-resistant to a particular antibiotic (Fig. 3d). The performance of these classifiers was evaluated on the testing partition.

Like strain classification, the prediction accuracy on both training and testing partitions increased with the embedding dimension. At $E = 10$, representing ~10-fold data compression (from the original dimension of $D = 98$ to $E = 10$), we achieved testing classification accuracies within 5% of using the raw data. For all four antibiotics, the low-dimensional classification accuracy achieves parity with using the uncompressed growth curves, at $E < 30$. For GM and SXT antibiotics, embeddings at $E = 10$ already exceeded the performance of the full data and the accuracy continued to increase until leveling off at $E = 30$ (Fig. 3e).

### Low dimensional embeddings enable efficient parameter estimation

In addition to classifying strain identity and antimicrobial resistance, we wondered if latent embeddings could be used for predicting quantitative properties. To this end, we constructed a parametrized ODE model of our bacterial isolates in our Group 2 dataset. For each one of these 311 bacterial isolates, we measured their growth dynamics in Lysogeny Broth (LB), LB with amoxicillin, and LB with amoxicillin and a $\beta$-lactamase inhibitor, clavulanic acid. Our ODE model thus incorporated kinetic parameters associated with both intrinsic growth dynamics and antibiotic responses (Methods).

We then constructed a machine learning pipeline to estimate the parameters associated with the ODE model, from experimental data (Fig. 4a). First, using the ODE model, we simulated system dynamics using 10,000 different parameter sets; in each combination, each parameter is drawn from a uniform distribution constrained to a biologically plausible range. For each parameter set, we simulated the growth dynamics with no antibiotic, with antibiotic, and with antibiotic and inhibitor concentrations corresponding to those used in experiments. We then concatenated the growth curves associated to these three conditions into a single data series. These simulated data were split into a 2:1 train/test split and the training set was used to train an autoencoder. To estimate parameters, we trained a multilayer perceptron (MLP) network to map the latent space to the ODE parameters used to generate the initial curves in the training set. Only the weights of the MLP were updated during training while the weights of the decoder were fixed. This practice ensured the combined NN made predictions only using the information encoded in the latent space learned during autoencoding.

After training, the performance of parameter estimation technique was assessed by applying our joint AE-MLP network on the test set, for which the ground truth was known. We observed that across training and test set, certain parameters could be estimated with high accuracy while others were much more challenging to estimate from raw training curves (Fig. 4b). Despite this, however, we found that using the predicted parameters to simulate ODE models yielded final predicted curves which captured the system final dynamics which high accuracy. This result is consistent with "sloppiness" of parameters associated with many dynamical models —large variation in their values yields little appreciable impact on the final dynamics[20,21]. Thus, the latent embeddings could be mapped to the 'stiff' parameters of the ODE model. We applied the trained AE-MLP model to the Group 2 dataset to estimate parameters that would allow the prediction of these data using the ODE model. Since the ground truth for the experimental system parameters is unknown, we assessed their quality by the quality of simulations generated by the estimated parameters from the AE-MLP model. We found that our ML pipeline with $E = 10$ generated parameters that enabled high quality growth curve predictions (Fig. 4c).

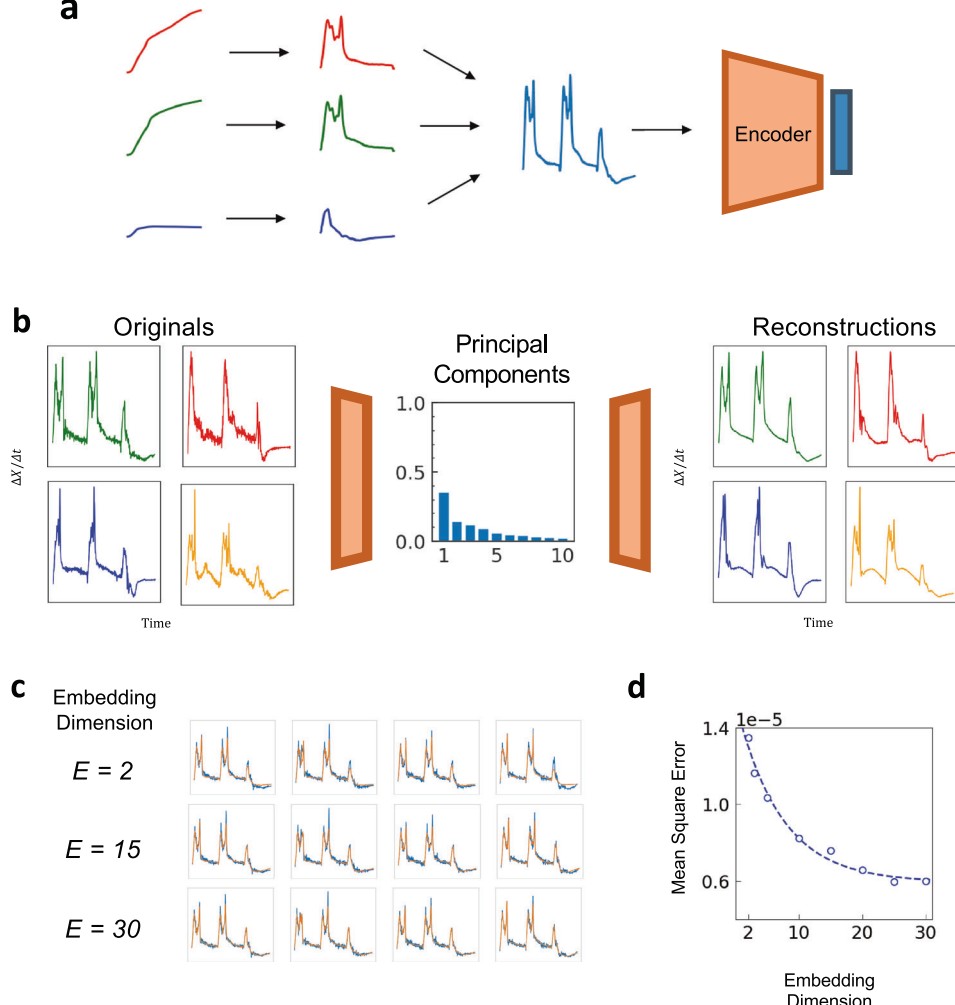

**Fig. 2 | Autoencoder compression of experimental bacterial growth curves.**
**a** Pre-processing growth curves for training. Each training example consisted of the finite difference of three separate growth curves corresponding to colonies of the bacterial strain grown in three different culture conditions (Supplementary Fig. 2). For a given training example, we took a time course for each growth curve conditioned and computed its time derivative. The three growth curve derivatives were concatenated into a single vector, which was then used to train the autoencoder. **b** Training an autoencoder using experimental data. An autoencoder was trained on 3732 time series generated from 311 clinical isolates with 12 examples per isolate and $E = 30$. Samples of original curves are shown on the left; the corresponding reconstructions are shown on the right. The center shows amount of variance explained by the first ten principal component of the latent space. The increased complexity of the growth curves is consistent with the use of multiple latent dimensions needed to generate high quality embeddings. **c** Reconstruction quality increases with $E$. The reconstructed curves (orange) better match the corresponding original curves (blue) as $E$ increases. With even small dimensionality the principal peaks and valleys of the curves are already captured in the reconstruction and with increasing embedding dimension, the reconstructions more closely fit to the smaller contours of the initial growth curve. **d** Reconstruction error decreases with $E$. The mean square error (MSE) between the initial growth curves and the reconstructed curves initially declines rapidly with increasing $E$ before leveling off as $E$ approaches 30.

## Predicting growth dynamics from initial conditions using embeddings

At its core, the analysis above demonstrates the ability of the autoencoder to dramatically compress high dimensional microbial dynamics while retaining sufficient dynamical information to enable phenotypical inference. Another common challenge is to predict the dynamics of a community starting from different initial conditions[16,22–24]. The ability to do so can guide predictive assembly of a microbial community to achieve desirable compositions and functions[5,15,16,23–27]. To this end, we wondered whether and to what extent a low-dimensional embedding can enable such predictions.

For this analysis, we used a specialized autoencoder, variational autoencoder (VAE), which embeds the learned representations into an $E$ dimensional, isotropic Gaussian distribution (Methods, Supplementary Fig. 3). This embedding ensures that the learned latent space is compact and continuous. This latter property ensures that interpolated new points in the latent space within the Gaussian distribution will approximate realistic trajectories in the phase space when decoded. This feature is critical for using the latent space to predict community dynamics starting from arbitrary initial conditions.

We simulated the growth dynamics of a two-member microbial community 6400 times, each with fixed parameters but starting from a different initial condition (Methods). Using a VAE, the growth curves can be visualized as trajectories in a two-dimensional phase space (Supplementary Fig. 4). Here, we compressed the growth curves of both species simultaneously into a single latent representation. Thus, each point in the latent space corresponds to a representation of an entire community's dynamics. We also used the full growth curves instead of their derivatives to prioritize representations that would directly correspond to trajectories in the system's phase space.

We used 4800 (out of 6400) growth-curve pairs as the training set. Even with $E = 2$, the trained autoencoder learned the dynamic flow of our system with high accuracy, as evidenced by the matching topologies of the reconstructed curves to our initial phase space

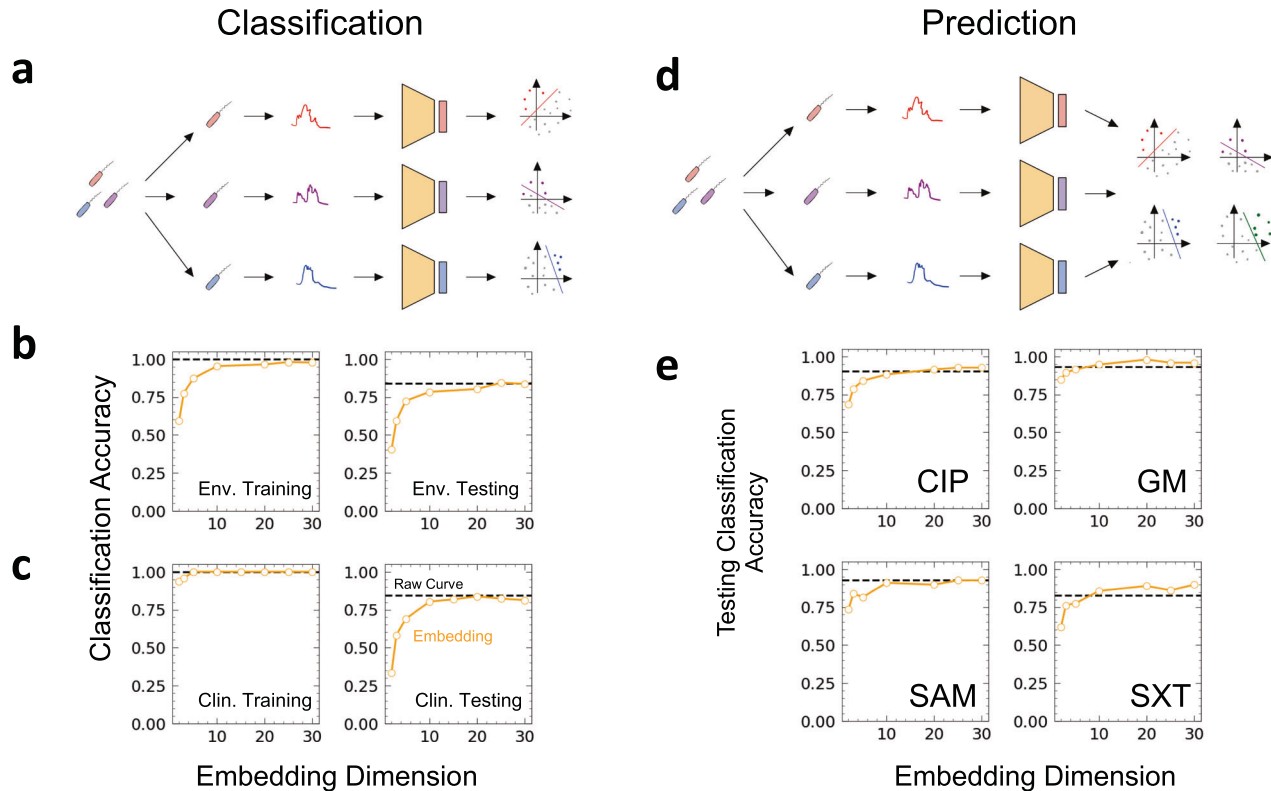

**Fig. 3 | Classifying bacterial phenotype using embeddings. a** Procedure for classifying growth curves using embeddings. We begin by culturing each strain to generate a set of growth curves. All growth curves are aggregated across all strains to train a single autoencoder. After the autoencoder is trained, the low dimension compression of each curve given by the encoder is used to generate a new dataset consisting of embeddings. $N$ separate support vector machine classifiers are then trained used to classify each strain using embedding. **b** Classification of environmental bacterial isolates. 12 growth curves were generated for each of 143 isolates (Methods); each growth curve consists of 98 time points. The classification accuracy (orange line) on both the training set (left) and the testing set (right) increases with $E$ before converging asymptotically to the classification accuracy achieved by using non-compressed curves (dashed line) around $E = 10$ to $E = 15$. **c** Classification of clinical bacterial isolates. 4 growth curves were generated for each of 244 isolates; each curve consists of 98 time points (Methods). The classification accuracy on training and testing sets rapidly increases with $E$ before saturating the accuracy achieved using the non-compressed curves (dashed line) around $E = 10$ to $E = 15$. **d** Procedure for predicting antibiotic resistance. A classifier is trained across all embeddings for each class of antibiotic response. Four classifiers were trained to predict whether a strain would be resistant to four different antibiotics: Ciprofloxacin (CIP), S-Adenosylmethionine (SAM), Gentamicin (GM), and Sulfamethoxazole (SXT). **e** Prediction accuracy for antibiotic resistance. 976 unique growth curves of 244 isolates with known resistances to one or more of the four antibiotics (SAM, GM, SXT, CIP) were compressed and used to train four SVMs. The prediction accuracy on the test set is shown for the four antibiotics. For CIP and GM, the embeddings at $E = 2$ performed at comparable accuracy as that using the non-compressed data.

(Supplementary Fig. 4b). The autoencoder seemed to represent the structure of the phase space in a form of local polar coordinates based around the fixed points of the dynamics. As the distance of a point in the latent space from the center of the Gaussian distribution increases, so does the arclength distance of the corresponding interpolated trajectory from the fixed point. For a fixed radius, the angular position of a point in the latent space relative to the mean of the Gaussian maps to the angular position of the initial condition of the matching reconstructed curve with respect to the fixed point (Supplementary Fig. 4c).

To predict growth dynamics from arbitrary initial conditions, we combined a multilayer perceptron (MLP) with the decoder component of the VAE. The MLP maps each initial condition into the learned latent space of the trained variational autoencoder. The decoder then maps the latent variables to growth curves, which would represent the prediction originating from the initial condition (Fig. 5a). During training, the combined neural network aims to optimize the parameters of the MLP encoder to minimize the mean square error between the neural network prediction and ground truth generated from numerical simulation.

We trained the VAE-MLP model using the same training set consisting of 4800 pairs of growth curves. We evaluated the performance of the trained model using the testing set consisting of 1600 pairs of growth curves. The predicted test set phase closely matched with the ground truth (Fig. 5b), evident in the strong agreement between predictions and ground truth ($R^2 = 0.998$). We repeated this same analysis using datasets generated from different two-member communities and from three-member and five-member communities. Across all these simulations, the hybrid VAE-MLP model performed consistently well ($R^2 = 0.990$) across test partitions (Fig. 5b). Comparing the entire phase spaces and individual microbial community growth dynamics examples confirms this result (Fig. 5b, c).

To further analyze the ability of the latent space to predict other complex dynamics, we considered two closely related tasks. One is to predict the dynamics from initial conditions of microbial communities that involve horizontal gene transfer. The other is to predict the dynamics of a microbial system that experiences transient perturbations. For the former, we simulated a two-member microbial community exchanging two plasmids. We found that the VAE-MLP pipeline could achieve high accuracy predictions using $E = 2$ (Supplementary Fig. 5). Interestingly, we could achieve this high-quality prediction with a VAE trained on only the species community dynamics, without explicit information being provided on the plasmid dynamics.

For temporal perturbations, we considered a three-member community that experiences a pulse of antibiotic treatment. We

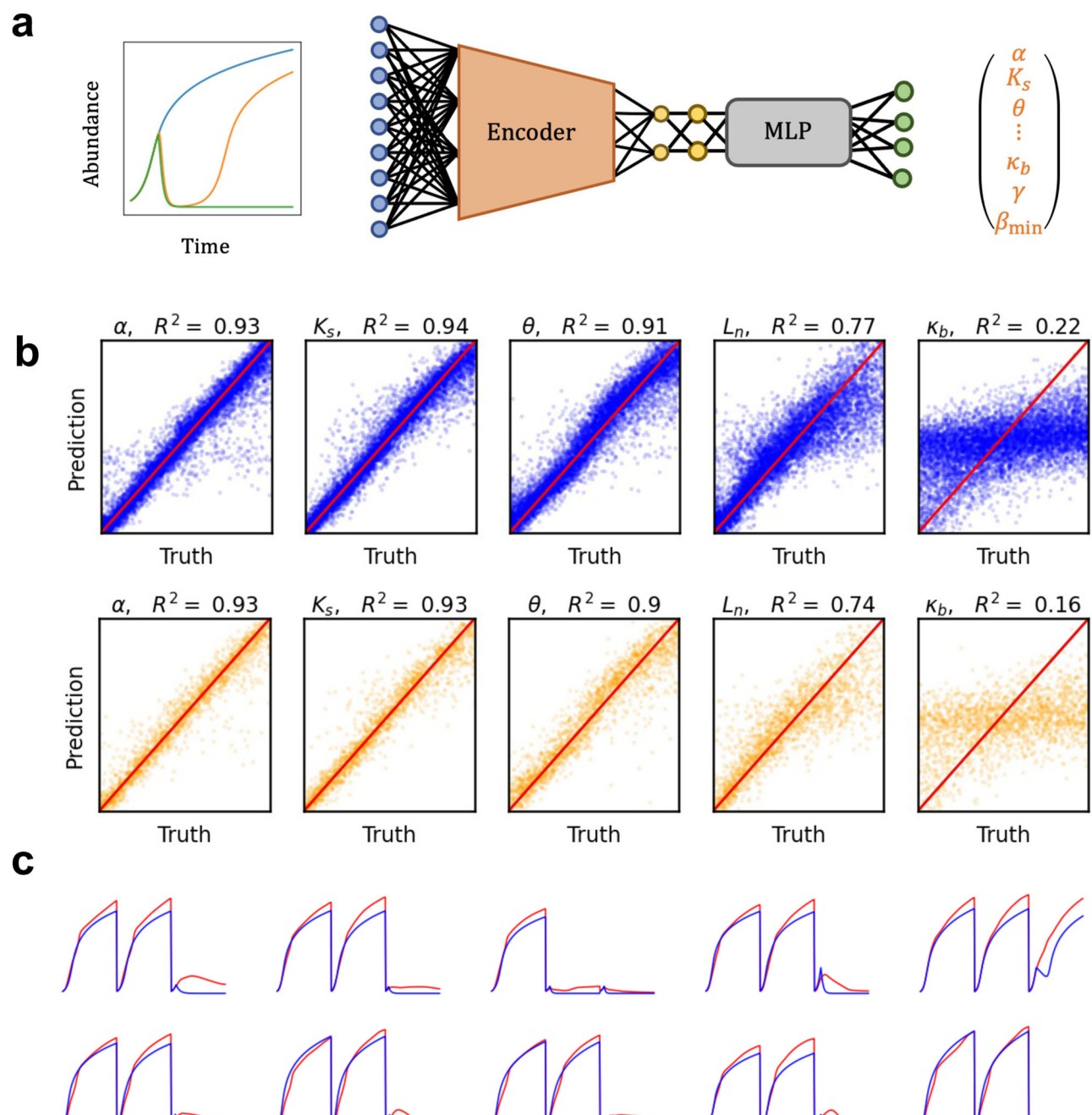

**Fig. 4 | VAE latent spaces can map to kinetic parameters of an ODE model.**
**a** Mapping from growth curves to parameters. The encoder from a trained VAE is used to map growth curves to a low dimensional latent space. A multilayer perceptron is then trained to estimate model parameters from the latent space. The VAE and MLP models are trained using simulated data with a wide range of growth parameters selected at random from a distribution over a biologically plausible range. The individual training examples correspond to three growth curves generated using an ODE model incorporating the effects of beta-lactam antibiotic and Bla-inhibitor as in Fig. 2 (Methods). Each curve corresponds to a different combination of initial concentrations of the antibiotic and *Bla* inhibitor. **b** The neural network enables accurate estimates of some but not all parameters. We show the comparison of five estimated parameters vs the ground for training (blue) and test (orange) sets of growth curves. The first four parameters can be estimated with higher accuracy. The fifth parameter, $\kappa_b$, is poorly estimated, suggesting it is a 'sloppy' parameter. **c** Estimated parameters generate accurate predictions of the growth dynamics. We apply our latent-space mapping procedure to experimental growth curves used in Fig. 2 (Supplementary Table 1). Despite being only trained on simulated growth curves, NN-estimated parameters from the experimental growth curves can enable the mechanistic model to predict growth curves with high fidelity.

generated 5000 sets of dynamics by the dose and time of the antibiotic pulse. We then used our MLP-VAE pipeline to predict community dynamics from the combinations of antibiotic dose and time. The pipeline enabled highly accurate predictions even at low embedding dimension ($E = 10$) (Supplementary Fig. 6).

## Predicting community dynamics from model parameters
Another common task is to predict the dynamics of a population while varying a set of system parameters, which is relevant for understanding or controlling the dynamics of a microbiome under different experimental conditions[5,26,28]. We thus examined if the autoencoder

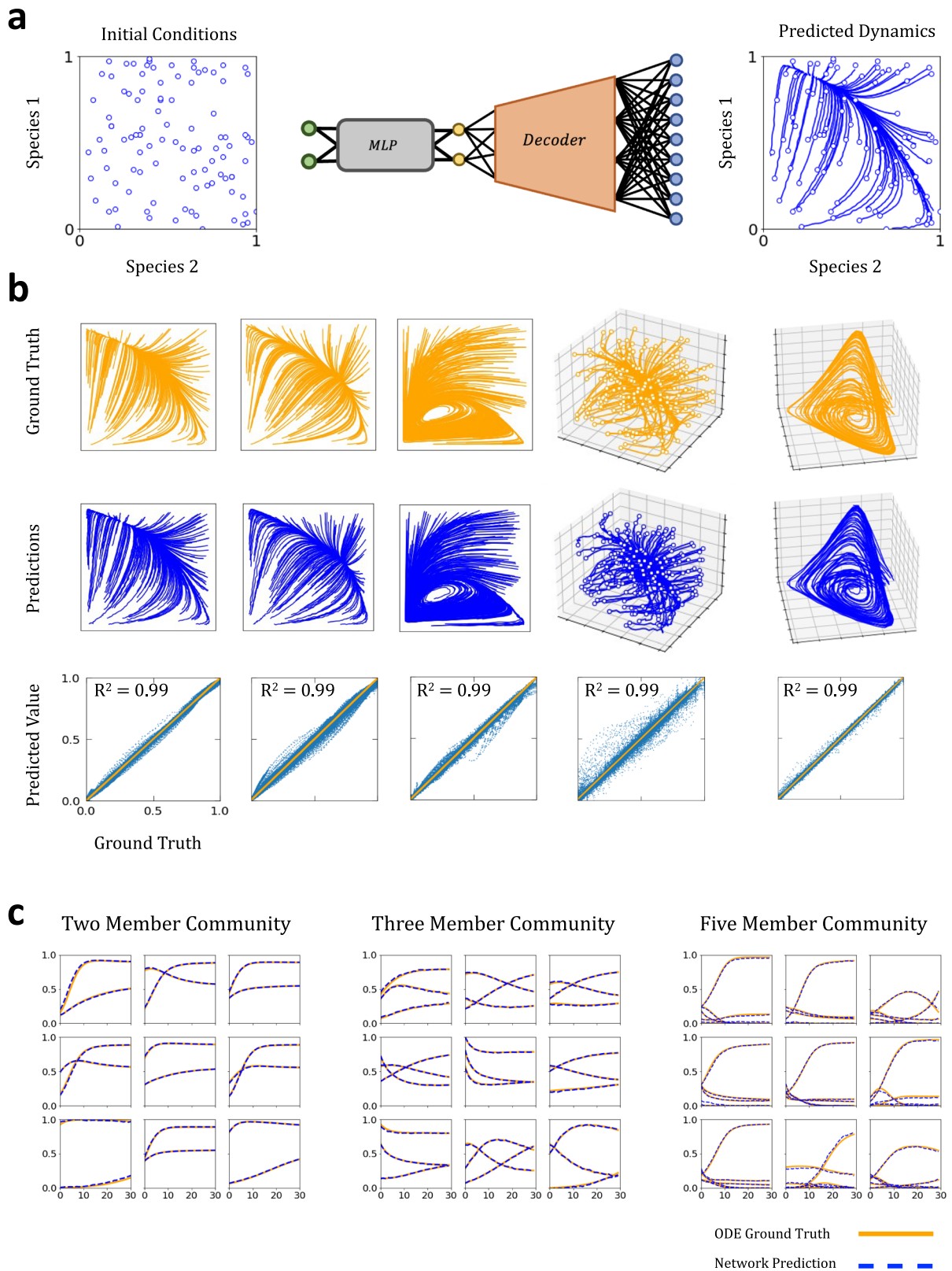

representations could be used to achieve parameter-to-dynamics predictions.

We used Eq. 2 to simulate dynamics of communities with increasing complexity ($N = 3, 4, 5,$ and 6) (Table 1). For each simulation, the initial abundance of each member was fixed to 0.1 arbitrary unit, but model parameters were sampled independently at random. Growth rates were sampled from the $Uni(0, 1)$ distribution while species-species interaction strengths were sampled from $Uni(-1, 1)$. We set the population self-interaction terms $\gamma_{ii} = 0$ so that intraspecies competition was captured by the growth rates. For each community size, we generated 10,000 simulated growth trajectories; each was split 3:1 into training and test sets. The training sets were used to train VAEs with varying embedding dimension $E$ (Methods).

**Fig. 5 | Predicting dynamics from initial conditions using VAE embedding.**
**a** Two-step initial condition to trajectory mapping. We combined the pre-trained decoder component of the VAE with an MLP encoder. The MLP encoder maps initial conditions of growth dynamics to the VAE embedding. The VAE decoder then translates the latent embedding into phase space to generate a predicted trajectory. During training, the parameters of the MLP encoder are optimized to minimize the mean square error between predicted trajectory and a ground truth generated from ODE simulation starting from the initial condition. The parameters of the VAE decoder are fixed during MLP training. **b** Accuracy of predicted dynamics. We trained several predictive models using different sets of simulated growth curves, each corresponding to different sets of gLV parameters and/or community sizes. For two-member communities, we trained our predictive model, both VAE and MLP components, using 4800 training examples and then evaluated their performance on 1600 testing examples with an embedding dimension $E = 2$. For the three-member and five-member communities, we used 3:1 train/test split for training and evaluation with $E = 3$ and $E = 5$ respectively (Methods). The top row shows a random subset of the simulated test set curves (in orange). The middle row shows the corresponding reconstructed curves (in blue). The bottom row shows a linear regression between predictions and the ground truth for all time points in all curves in the test set. Perfect alignment corresponds to the line $y = x$. The MLP-VAE models achieve high quality predictions ($R^2 \geq 0.99$) for ecological dynamics with fixed point (columns 1,2, and 4), limit cycle (column 3), and chaotic attractors (column 5). **c** Sample predicted dynamics for two, three, and five member communities. Growth curves predictions (dashed blue) for two-member, three-member, and five-member communities and the corresponding ground truth curves (solid orange). For five-member communities simulated, the average $R^2$ was 0.998.

To achieve parameter-to-dynamics prediction, we used an MLP to map ODE parameters to the embeddings. The trained decoder then maps the embeddings to growth dynamics (Fig. 6a). During training, the weights of the pre-trained decoder were frozen so that the system should construct predictions based on the latent embeddings and information learned during autoencoding.

For each community size, we trained the MLP encoder using the same training set used to train the VAE. We then assessed the quality of model parameter-to-dynamics prediction based on performance on the test set. Figure 6b shows the results of this analysis for a three-member community. As $E$ increases, there is an initially rapid increase in qualitative prediction quality before saturating after a critical $E^*$. This is evidenced by the rapid improvement both qualitatively in test-set predictions visualized in phase space, and quantitatively increasing $R^2$ coefficient between ground truth and predicted values. Rapid improvement occurs between $E = 3$ to 6 but relatively smaller improvement between $E = 6$ to $E = 9$. That is, $E^* < N^2$. Despite the ODE model requiring $N^2$ unique parameters to successfully simulate different growth curves, the VAE model learns a smaller set of "latent parameters" which can also be used to uniquely represent each community and predict their dynamics with high fidelity.

Repeating this analysis for $N = 4, 5$, and 6 member communities while maintaining the neural network architecture consistent revealed a similar trend: the quality of reconstruction initially improves rapidly before saturating, after which prediction quality saturates. The mean square prediction error drastically drops before plateauing after a threshold $E$ value ($E^*$) (Fig. 6C), which can be determined by a sufficiently small threshold error ($1.5 \times 10^{-7}$). $E^*$ increases with size of the initial community, reflecting the need for a larger latent dimension for a larger community. However, as illustrated in Fig. 6D, $E^*$ increases approximately linearly with the community size. In contrast, the number of ODE parameters increases as the square of the community size. That is, the number of effective parameters used by the autoencoder for predicting dynamics for a fixed initial condition scales much more slowly than what is needed to parameterize a standard mechanistic model.

### Predicting experimental community dynamics from system parameters

To experimentally test our ML pipeline, we generated a dataset of 7200 time courses corresponding to triplicate measurements of GFP and OD of 1200 configurations of two-member microbial communities (Methods): (12 strain/plasmid/drug combinations) × (100 combinations of drug concentrations for each strain/plasmid/drug combination). Each of these communities consisted of one member that carried a plasmid which conferred both fluorescence and antibiotic resistance and another lacking both fluorescence and resistance. During experiments, we measured the total OD and the GFP fluorescence for 145 time points over 24 h as surrogate measures of the community dynamics.

We split our data 2:1, with 4800 curves in our training set and 2400 in our test set. The training data was used to train a VAE with varying latent dimensions. We then trained a MLP to map the experimental configuration (strain/plasmid/drug combination/drug concentration) to the latent space, which was decoded, using the decoder of the trained VAE, to community dynamics (Fig. 7a). Even with $E = 5$, which is much smaller than the initial dimension of the concatenated data ($D = 290$), the trained MLP-VAE enabled high-accuracy prediction of both GFP and OD dynamics (Fig. 7b). The same VAE latent spaces also allowed classification of community growth curves to match experimental configurations (strain/plasmid/drug combination) (Supplementary Fig. 7).

## Discussion

Here we show that autoencoders can map dynamics of microbial communities into a low-dimensional latent space that contains sufficient information to enable study of key biological properties, such as determination of strain identity, antibiotic resistance, or predicting dynamical trajectories. In other words, the high dimensional growth curves possess a low-dimensional "latent structure" that ML models can exploit for downstream analysis.

An immediate practical application is the use of autoencoder for building pipelines for microbial dynamics analysis. The autoencoder embeddings provide a non-biased approach to denoising growth curves without losing critical biological information. The level of detail to be maintained can be tuned by choosing the embedding dimension. In addition, mapping growth data to a low-dimensional space could enable simpler, more efficient, and more effective model optimization, as demonstrated in the use of embeddings for strain classification and resistance prediction. Furthermore, the latent space of VAE models serves as a continuous, compact approximation of the underlying distribution of growth curves which can be used for downstream machine learning tasks, including generative tasks such as predicting growth dynamics from parameter sets or initial conditions. Already, exploiting the "latent structure" of datasets through such regularized autoencoder representations is the critical first stage for most modern ML techniques spanning audio processing, single-cell multi-omics, and computer vision[29–36]. Here we demonstrated that these representation learning methods hold similar promise for the study and engineering of microbial community dynamics.

The existence of a low-dimension latent structure in microbial community dynamics has implications for effective modeling and measurements of such dynamics. For a system that is sufficiently well understood, a mechanistic model represents a much more concise representation of the system than the raw data. For instance, if population growth can be well described by the logistic model, only two parameters (growth rate and carrying capacity) are needed for predicting growth, whereas a sufficiently high-resolution experimental sampling is needed to generate a high-quality growth curve. That is, the mechanistic model represents an effective dimension-reduction,

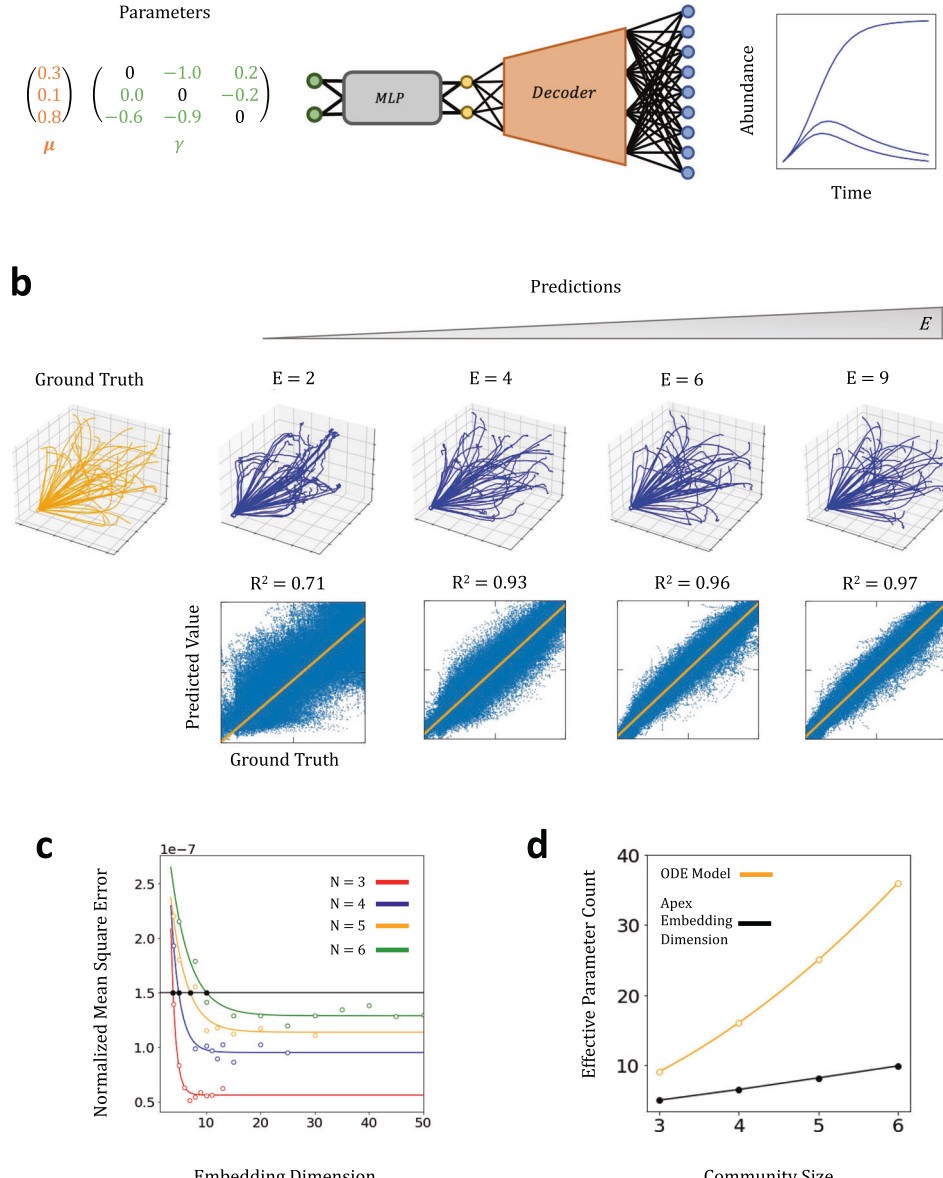

**Fig. 6 | Predicting dynamics from system parameters using VAE embedding.**
**a** Two-step parameter to trajectory mapping. For an $N$-member community, the $N^2$ parameters for the ODE model are mapped via a MLP encoder to the pretrained VAE embeddings. The VAE decoder then maps these embeddings to predicted growth curves. During MLP training, the weights of the VAE decoder are frozen; only the weights of the MLP encoder are tuned to minimize the error between the predicted dynamics and ground truth dynamics. **b** Prediction quality initially increases before saturating with embedding dimension. The predicted dynamics in phase space for one hundred communities drawn randomly from the test set partition are shown for increasing embedding dimension. Increasing $E$ from 2 to 6 led to substantial qualitative and quantitative improvement in the prediction accuracy, as evidenced by phase space visualization and $R^2$ coefficients respectively. Increasing $E$ from 6 to 9 led to more incremental, suggesting that past some intrinsic or "apex" maximum

embedding dimension, the quality of prediction saturates. **c** Normalized mean square error (MSE) decreases with the embedding dimension. Normalized MSE is calculated by computing the MSE of the predicted dynamics from the ground truth value and then normalizing by the average square norm of the ground truth dataset. For each community size ($N$), the error vs embedding dimension is fit with an exponential curve. The MSE drops rapidly before saturating at $E^*$, which corresponds to a sufficiently small threshold error ($1.5 \times 10^{-7}$). $E^*$ indicates the number of parameters needed to reconstruct the community dynamics with the pre-specified fidelity; it increases with $N$. **d** The critical dimension $\left(E^*\right)$ grows more slowly than ODE parameter count. As the size of the community ($N$) increases, the number of unique parameters needed for the ODE model scales $O\left(N^2\right)$, but $E^*$ scales approximately $O(N)$.

compared to the direct representation of the system dynamics using time-course data.

However, a mechanistic model, even if perfectly accurate, can suffer from the "curse of dimensionality" as a system gets more complicated. For a gLV model, the number of parameters grow on order $O(N^2)$ with the number of species; for models that also incorporate plasmid growth the number of parameters can grow exponentially[37,38]. Our results indicate, however, the community dynamics can be

represented using many fewer variables than the number of parameters in the mechanistic model. That is, the effective dimension of system can be much smaller than the number of parameters needed for a mechanistic model.

This compressibility is a manifestation of the "sloppiness" of many dynamical models: a specific system output often depends on only a minority of the parameters (stiff parameters) but insensitive to changes in other parameters (sloppy parameters)[21,39]. The embedding

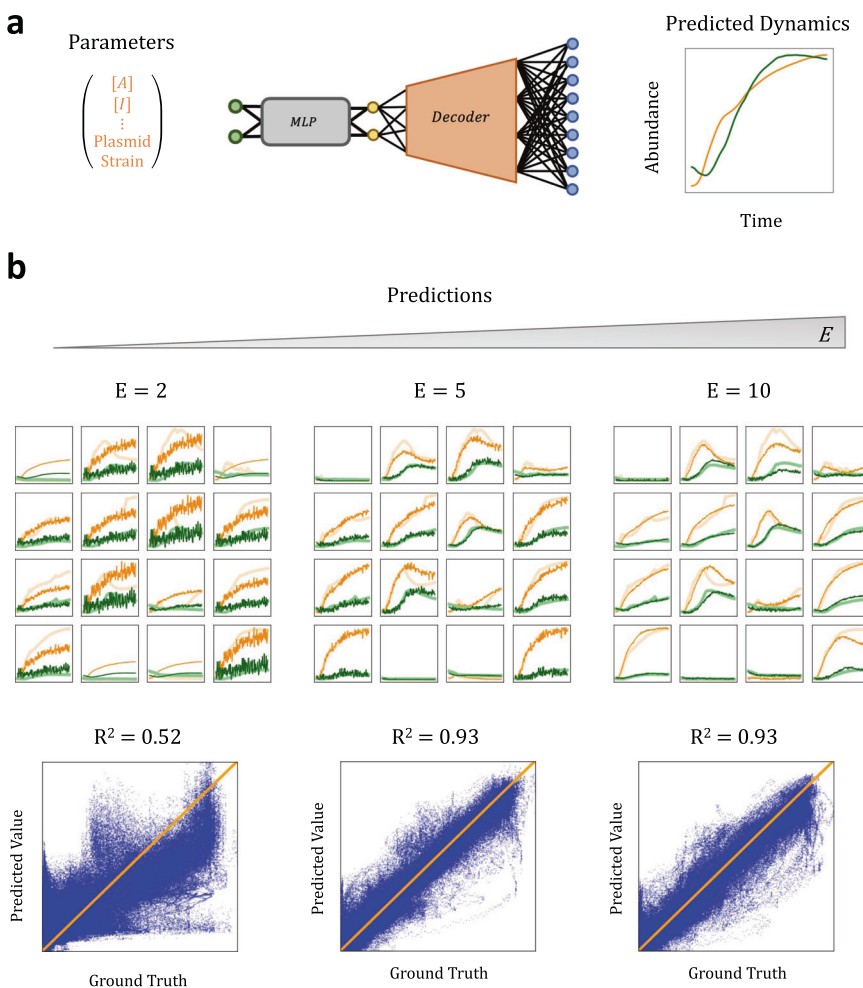

**Fig. 7 | VAE enables dynamics prediction for two-member communities. a** Two-step growth curve to parameter mapping. Our procedure for predicting the growth dynamics from growth parameter for two-member microbial communities is identical for simulated curves. For parameters, we consider the *E. coli* strain type, antibiotic resistance plasmid type, initial antibiotic concentration, inhibitor type, and initial inhibitor concentration (Methods). We aim to map these to the growth dynamics of the two-member community consisting of one strain carrying antibiotic resistance genes and one sensitive strain. Only the resistant strains produce GFP, so the total optical density (OD) measurement (orange) can be treated as a measure of the total community biomass while the GFP intensity (green) is a surrogate measure of the resistant strain. **b** Sample community predictions for various embedding dimensions. We plot the predictions (thin line) vs ground truth (thick lines) for randomly chosen communities. As the embedding dimension of the embedding increases, the resultant quality of predictions also improves. This is corroborated by plotting the predicted vs ground truth values for the entire dataset in 2D cartesian grid. As the embedding dimension increases, the points cluster closer to the line and accuracy increasing is corroborated with an increasing $R^2$ value. We note the accuracy dramatically improves between E = 2 to E = 5 but saturates at E = 10.

variables from an autoencoder may represent the combination or transformation of the stiff parameters. As such, the autoencoder compression can be used to estimate the number of effective parameters necessary to capture the dynamics of interest (e.g., growth curves). In our analysis, the classification accuracy of distinguishing resistant from non-resistant strains plateaus at $E \geq 10$. This implies that around 10 parameters would be sufficient to constrain a model to predict resistance.

Nevertheless, this compressibility can explain the effectiveness of other types of coarse-graining and abstraction, such as the development of a simple metric for predicting plasmid persistence in a microbial community[37] or basic principle for understanding mutualistic communities[40]. A central theme across these studies is that though quantitative microbial models are often complex, they frequently give rise to simpler rules that emerge from structural redundancy[20,41]. This work provides additional evidence, discovered directly from data, of this emergent simplicity in microbial systems. This property suggests that the analysis of complex microbial communities should focus on identifying the essential variables for both experimental design and modeling analysis, instead of attempting to model and measure every observable output. While we focus on microbial community dynamics, our approach and conclusion could be applicable for the analysis other high-dimensional biological systems, such as intracellular gene expression[42–44], cancer development[45,46], and macro-scale ecological interactions[2,47].

Several caveats exist, however, for using low dimensional autoencoder embeddings to analyze microbial growth dynamics. Though autoencoders can be used to probe the sloppiness of parametrized mechanistic models of microbial growth as well as develop general purpose representations of microbial dynamics, the embeddings they generate themselves are not directly biologically interpretable. The values of the latent variables are constrained by several factors associated with neural network training, including: the neural network architecture and parameterization, the size of the latent space, as well as the hyperparameters during training (e.g., batch size and learn rate). Two different autoencoders with same latent dimension size can achieve similar reconstruction fidelity and performance on downstream prediction and classification tasks while using very different

sets of latent variables. However, once they are deduced, the latent embeddings can be mapped to variables and conditions that have concrete interpretations. For instance, we have demonstrated that the latent embeddings can be mapped to the initial conditions, kinetic model parameters, and experimental configurations. On the technical front, this mapping can be more effective than direct use of the original, uncompressed data, as evidenced by the improved performance in strain ID and resistance prediction using the latent embeddings. On the conceptual front, the mapping provides an indirect manner to interpret the latent variables, through the parameters or system configurations they are mapped to. Furthermore, in this study we use various canonical architectures of autoencoder and other neural networks as a tool to demonstrate that despite the apparent complexity of microbial dynamics and communities, the number of effective variables needed to describe them can be very small. In the future work, it would be interesting to evaluate and benchmark various autoencoder architectures to generate the most informative representations for microbial community dynamics.

## Methods

### Terminology

Throughout this report, we discuss the dimensionality of the growth curves, latent spaces, microbial communities, and parameter spaces. We summarize these below with examples:

### Numerical simulation of community dynamics

Population and community growth curves were simulated with variations of the logistic growth and generalized Lotka-Volterra models. For the initial one-member communities shown in Fig. 1a, we modeled the system using the logistic growth equation (Eq. 1), where the growth rate parameter was sampled from a Gaussian distribution with mean $\mu = 0.13$ and standard deviation $\sigma = 0.02$. We simulated in total 1000 growth curves, each for 101 equally sized time points over the time interval $[0, 100]$ total time units for a population starting with a relative abundance of 0.05. The derivative of the growth curve was estimated by computing the finite difference between adjacent time points, yielding 100 time point final curves. These were then used as input to the autoencoder.

The community simulations in Fig. 1b, c were generated using a modified gLV (Eq2):[8]

$$\frac{dp_i}{dt} = \mu_i p_i \left( 1 - p_i - \frac{\sigma}{1 + \sum p_j \gamma_{ij}^+} - \sum p_j \gamma_{ij}^- \right) \quad (2)$$

Here $p_i$ corresponds to the relative abundance of species $i$, $\mu_i$ is the specific growth rate of species $i$, $\gamma_{ij}$ represents the influence that species $j$ has on species $i$, and $\sigma$ represents background stress of the environment on species growth rate. We treat separately the positive interactions, $\gamma_{ij}^+ > 0$, in the denominator where they increase the rate of growth by reducing background stress. Negative interactions, $\gamma_{ij}^- > 0$, on the other hand directly subtract from the total growth rate of the species proportional to the abundance of inhibitory species. This formulation ensures that the relative abundance of all species does not ever grow unbounded to infinity.

In these simulations, growth rate for each species was drawn at random from a Gaussian distribution with mean $\mu = 0.10$ and standard deviation $\sigma = 0.02$. Entries in the community interaction matrix $\gamma_{ij}$ were similarly all sampled identically and independently (i.i.d.) at random with $\mu = 0$ and standard deviation $\sigma = 1.0$. The stress parameter $\sigma$ was fixed across all simulations at 0.05. We simulated in total 1900 different five member communities, yielding in total 9500 unique growth curves, with each species starting with a relative abundance of 0.05. Each was numerically integrated for 134 timesteps on the interval over the time interval of $[0, 1000]$. The finite difference method was

used identically as with the single species population growth simulations to estimate the growth curve derivative.

### Simulation of microbial dynamics with antibiotic and antibiotic response

To model the dynamics of our microbial strains of our Group 2 dataset with and without antibiotic and antibiotic inhibitor, we developed the following model of three coupled odes:

$$\begin{aligned} \frac{dn}{dt} &= (\alpha g - \beta \ell)\, n \\ \frac{db}{dt} &= \beta \ell n - D_b b \\ \frac{da}{dt} &= -\left(\kappa_b b + \phi n\right)\frac{a}{K_a + a} - d_a a \end{aligned} \quad (3)$$

Where $n$ is the cell density, $b$ is the concentration of extra-cellular $\beta$-lactamase, and $a$ is the concentration of beta-lactam antibiotic. Starting, with $n$, the size of the population is governed by growth due to nutrient consumption and cell division as captured by $\alpha g$ and death due to antibiotic induced lysis as captured by $\beta \ell$. $\alpha$ represents the maximum population growth rate, but it is modulated by $g$ to capture the complexity of experimental data:

$$g = \left( \frac{1}{1 + \left(\frac{n}{N_m K_s}\right)^\theta} \right) \left( 1 - \frac{n}{N_m} \right)$$

Here, $N_m$ is the carrying capacity of the population. $K_s$ and $\theta$ are shape parameters for the growth curve. The death term is given by the product of $\beta$, constrained between 0 and 1, which measures the susceptibility of an individual population to $\beta$-lactam antibiotic and $\ell$, the lysis rate as determined by the concentration of antibiotic ($a$). We define $\beta$ as:

$$\beta = \beta_{\min} + c\,(1 - \beta_{\min})$$

$\beta_{\min}$ expresses the minimum susceptibility of an individual bacterium to $\beta$-lactam cytotoxicity. $c$, also constrained between 0 and 1, measures the permeability of an individual cell to $\beta$-lactamase inhibitor such as clavulanic acid. $c = 1$ implies a cell is totally permeable to antibiotic. $c = 0$ means a cell is totally impermeable to beta-lactamase inhibitor. If an inhibitor is present and $c > 0$, then we model the resulting inhibition of intra-cellular $\beta$-lactamase as an increase of the susceptibility of the population, with at most $\beta = 1$ for $c = 1$.

The lysis rate $\ell$ in turn is given by:

$$\ell = (\gamma g)\left(\frac{a^{h_a}}{1 + a^{h_a}}\right)$$

where $\gamma$ is the maximum lysis rate and $g$ is as defined above. $h_a$ is Hill coefficient. The antibiotic concentration ($a$) is scaled with respect to the half-maximum concentration, such that $\ell$ reaches half maximum when $a = 1$.

During simulations, we take $\ell = 0$, until the population size reaches some critical density $n > L_n$, after which $\ell$ is determined by the equation above for the remainder of the simulation even if $n$ falls below $L_n$. This constraint captures the fact that there is time-lag between the initial exposure to antibiotic where individual bacteria need to accumulate damage to lyse[48].

For extra-cellular beta-lactamase, the dynamics are primarily driven by the release of intracellular beta-lactamase from lysing cells, as captured by $\beta \ell n$ term, and a basal decay rate of $D_b$.

Lastly, for the antibiotic, the dynamics are driven by degradation by both extracellular and intracellular beta-lactamase as well as a basal

decay in solution. Degradation by $\beta$-lactamase is captured in the term:

$$-(\kappa_b b + \phi n)\frac{a}{K_a + a}$$

Where $\kappa_b$ is the degradation rate by extracellular beta-lactamase. $\phi$ in turn captures the degradation rate due to intracellular beta-lactamase and is thus proportional to the microbial population. We calculate $\phi$ according to:

$$\phi = \phi_{max}(1 - c)$$

Where $\phi_{max}$ is the maximal degradation rate due to intracellular beta-lactamase. $c$ is our previous permeability to beta-lactamase inhibitor. For totally permeable cells, $c = 1$, implying that the inhibitor totally blocks all intra-cellular beta-lactamase activity. The final Michealis-Menton term ensures that degradation occurs in an antibiotic density dependent manner. The $d_a$ similarly captures the background decay of the antibiotic in solution not due to $\beta$-lactamase activity.

To generate the dataset of numerically simulated curves used in our VAE-MLP prediction pipeline, we generated random values for $\alpha, K_s, \theta, L_n, \kappa_b, \phi_{max}, \gamma, \beta_{min}, d_b$, and $c$ from truncated normal distributions constrained to a biologically plausible range. Detailed numerical values for these ranges can be found in the linked GitHub repository. For each set of parameters, we generated three distinct growth curves corresponding to different initial values of antibiotic and antibiotic inhibitor. Specifically, we generated a curve for the case of no antibiotic, the case of $a_0 = 10$ with no inhibitor, and $a_0 = 10$ with inhibitor. For each of these simulations, we integrated the system for 145 time points over a range of 24 hours, matching our experimental growth curves in group 2. The initial value of $n_0$ was drawn similarly from a truncated normal distribution with mean $\mu = 0.04$ O.D. and standard deviation $\sigma = 0.005$ O.D.

### Predicting community dynamics from different initial conditions

For our two-member communities, 6400 curves were generated using Eq. 2 for a given set of model parameters $\mu_i$ and $\gamma_{ij}$. $\sigma$ as with Figure1B and1C was fixed to 0.05. These model parameters were initially generated by sampling growth rate values $\mu_i$ uniformly between [0,1] and interaction parameters $\gamma_{ij}$ from the standard normal distribution. For our two-member communities, we integrated the system for thirty total steps between 0- and 20-time units, yielding 30-dimensional growth curves. No finite differencing was applied. Initial conditions were sampled in a square, equally spaced grid on the unit cube of $[0,1] \times [0,1]$. For three member communities, the same process was repeated except the simulation was run for 30 time points between 0- and 20-time units to enable most of the trajectories to reach steady state and we sampled 3375 total communities, this time with initial conditions on an equally spaced grid in the unit cube. For five member communities, the process was again identical except we ran the numerical simulation for 40-time units and sampled 7776 total initial conditions equally spaced on the five-dimensional unit hypercube.

To simulate dynamics corresponding to the 2D limit cycle shown in Fig. 5b, we used a modification on the standard generalized Lotka-Volterra incorporating nonlinear competition terms:

$$\frac{dp_1}{dt} = p_1(1 - p_1) - \frac{a\,p_2}{d + p_1} \qquad \frac{dp_2}{dt} = b\,p_2\left(1 - \frac{p_2}{p_1}\right) \qquad (4)$$

With $a = 1, d = 0.1$, and $b = 0.2$. As with the fixed-point dynamics, we sampled initial conditions at equal spacing from the unit square, in total sampling 5625 growth curves integrated between 0 and 20-time units. Here we sampled $D = 40$ total time points on this interval.

For the chaotic attractor, we utilized a previously studied variation of gLV and associated set of parameters which are known to generate chaotic dynamics[49]:

$$\dot{p}_i = p_i\left(\sum_j \gamma_{ij}\left(1 - p_j\right)\right) \qquad (5)$$

With:

$$\gamma = \begin{pmatrix} 0.5 & 0.5 & 0.1 \\ -0.5 & -0.1 & 0.1 \\ 1.43 & 0.1 & 0.1 \end{pmatrix}$$

We sampled points 8000 total initial conditions for these dynamics uniformly spaced on the domain $[0.3, 1.0]^3$ and ran all simulations on the time interval [0,20] for 30 time points.

### Predicting temporal dynamics from different parameter sets

For each community size, 10,000 communities were sampled of which 7500 were used in the training set and 2500 used in the test set, once again using Eq. 2. For each community simulation, the initial abundance of each community member was fixed to 0.1 arbitrary units but the growth rate parameters $\mu_i$ were sampled i.i.d. from the uniform distribution $Uni(0,1)$ and the elements of the interaction matrix $\gamma_{ij}$ were sampled i.i.d. from the uniform distribution $Uni(-1,1)$. We set the population self-interaction terms $\gamma_{ii} = 0$. Each growth curve corresponding to a species' growth dynamics within each community consists of 20 time points sampled on the time interval [0,20].

All numerical simulation was performed in Python leveraging the SciPy integration library with LSODA solver.[16]

### Causal convoluted autoencoder compression

For analysis associated for Figs. 1 through 3, we used a causal convolutional autoencoder for generating our low dimensional time series embeddings. The encoder network operates by iteratively applying one dimensional diluted causal convolution operations in each layer followed by ReLU activation functions, allowing for the encoder to generate representations that consider multiple different time scales of the growth curve (or concatenated growth curves) sample. The decoder consists of a standard multilayer perceptron which maps its input by a series of dimensionality increasing linear transformations followed by nonlinear ReLU activations. A schematic of these operations can be found in Supplementary Fig. 1. The choice of this encoder architecture was inspired by previous success in generating high quality latent representations, particularly by Francheschi et al.[50], from whom we derived the initial code for the encoder, and WaveNet[51].

During training, the network aims to minimize the reconstruction loss which is given by the mean square error between the initial curve and the network's reconstructions averaged across all training examples:

$$L_{recon} = \frac{1}{n}\sum_i^n |\mathbf{x_i} - \mathbf{x}_{recon}|_2^2$$

During training, we iteratively tune the model's internal parameters in both the convolutional and linear layers via the stochastic gradient descent. The number of total training iterations for each experiment to ensure that the network trained until this loss saturated (ranging generally from 5000 to 50,000 epochs). Additionally, across experiments we manually tuned the depth of the network, the number of convolutional layers, and the number of channels per layer hyperparameters to ensure that for a given dataset, simulated or experimental, the final mean square error converged and that the final reconstructions qualitatively captured the dynamics of each growth

curves such as capturing smaller peaks or valleys and matching curvature with the initial curves.

We found that swapping the diluted causal convolutions with more standard 1D convolutions did not affect the reconstructed quality of the simulated data.

All optimization was performed using the Adam optimizer and the neural network was implemented in python using the PyTorch version 2.0.0[20].

## Variational autoencoder compression

For dynamics prediction problems, we leveraged a variational autoencoder to ensure a continuous latent space to assist in generative tasks. Specifically, we use a convolutional variational autoencoder (CVAE), which applies iterative one-dimensional convolution operations in the encoder network to generate time-series specific features, which are then passed through a linear layer to generate a final latent embedding for each individual community. A more detailed look of our CVAE architecture is found in Supplementary Fig. 3. In the encoder portion of the network, these convolution operations extract features from the input community dynamics $x_i$ which in the final stage of the encoder are flattened into a single vector. These flattened features are then transformed by two parallel series of linear layers to two latent vectors, dubbed the mean vector $\boldsymbol{\mu}$ and standard deviation vector $\boldsymbol{\sigma}$.

During optimization, the VAE does not generate a single encoding vector $z_i$ for each input like a vanilla AE, but rather to each point associated a small region of the output latent space characterized by a Gaussian distribution with center $\boldsymbol{\mu}$ and standard deviation $\boldsymbol{\sigma}$. During training, the forward pass of the model deterministically generates a unique set of $(\boldsymbol{\mu}, \boldsymbol{\sigma})$ for each input datapoint $x_i$ (Supplementary Figure 3). The actual latent embedding is then generated by sampling a vector from this learned distribution via $\mathbf{l} = \boldsymbol{\mu} + \boldsymbol{\sigma}\boldsymbol{\epsilon}$ where $\boldsymbol{\epsilon}_i \sim N(0,1)$ random variable and then passing that sampled vector through a single dense linear (affine) layer $\mathbf{z} = \text{ReLU}(\mathbf{Wl} + \mathbf{b})$. This latent embedding is then decoded via the decoder as with a standard AE, with the prediction $\mathbf{x_{recon}} = \psi(\mathbf{z})$, which are then used, during training, to compute reconstruction loss as usual. This stochastic sampling encourages the neural network to not simply associate a single point with a particular training example but rather a small continuous region of the latent space. This in turn ensures the *local* region of the latent space near the mean vector $\boldsymbol{\mu}$ for each training example can be decoded and yield biologically meaningful interpolations.

To ensure that the variational autoencoder learns not just a *locally* continuous, but also a *globally* continuous distribution of latent embeddings, the VAE loss also incorporates an additional Kullbeck-Leibler divergence loss which measures the discrepancy of each latent embeddings form a standard Gaussian distribution:

$$L_{KL}(\mathbf{x_i}) = \frac{1}{2}\left(\sum_i^E \mu_i^2 + \sum_i^E \sigma_i^2\right) - \frac{1}{2}\sum_i^E (\log(\sigma_i^2) + 1) \tag{6}$$

Where $\mu_i$ and $\sigma_i$ are components of the latent $E$-dimensional mean and standard deviation vectors generated by the output of the encoder network for a given $\mathbf{x_i}$. This additional loss encourages that the regions of the latent space associated with training example cluster closely near one another at the origin and that region of the latent space associated with each training example do not grow individually too large. This causes the that the local regions associated with each embedding to overlap each other and overall ensure that the entire distribution *globally* remains continuous such that interpolating within the convex hull of the entire latent distribution remains biologically meaningful.

A more detailed discussion of the derivation and theory of this loss and greater architecture of the VAE can be found in Kingma and Welling (2019)[52].

Additionally, we found that adding an reconstruction loss which penalizes the model for errors on the first time point during reconstruction improved the ability of the curve to encode and then reconstruct the earlier transient phase of a given phase space trajectory. We dubbed this the boundary point loss (BPL):

$$L_{BPL}(\boldsymbol{x_i}) = \frac{1}{n}\sum_i^n |\mathbf{x_i}(0) - \mathbf{x_{recon}}(0)|_2^2 \tag{7}$$

where $\mathbf{x_i}(t)$ gives the point on the phase space trajectory at a given time $t$. The final loss of the model was then given by:

$$L_{total} = L_{recon} + \alpha L_{KL} + \gamma L_{BPL}$$

The hyperparameters $\alpha$ and $\gamma$ are used to vary the relative importance of KL and BPL penalties relative to the reconstruction loss and can be tuned up to place higher priority on either regularizing the latent space into a Gaussian or reconstructing the initial time point, respectively. During our numerical experiments we set $\alpha = 0.001$ and increased $\gamma$ between 0 and 10 from dataset to dataset until reconstructions stopped improving.

All optimization was performed using the Adam optimizer and the neural network was implemented in python using the PyTorch library[53].

## Classification of strains using support vector machines

Support vector machines are a class of binary linear classifier. They operate by learning an optimal hyperplane that separates two classes of data points. They can be used to separate nonlinearly separable datasets by using appropriate "kernels"[54].

For multiclass strain identity classification and antibiotic prediction experiments, we leverage SVM classifiers trained in a one-vs-all procedure. In our analysis, for each unique strain label or antibiotic resistance label one SVM classifier was trained to discriminate that strain vs all others such that for a dataset with $n$ classes a total of $n$ SVMs were trained. During training, the hyperparameters of each SVM were optimized via three-fold-cross validation on the training set, performing a grid-search through the hyperparameter space of $C$, the margin of error of the SVM, kernel type, either linear or RBF kernel, and $\gamma$, a parameter which tunes the relative "sharpness" of data point peaks for RBF kernel SVMs. A more detailed explanation and theory of these hyperparameters is provided in Hoffman[54]. During three-fold-cross validation, we partition the training dataset into three smaller sets and use two components for training and the third for testing the accuracy of a given hyperparameter configuration. After testing every hyperparameter set on a particular hold-out set, this validation set is swapped out with one of the other two partitions and the whole procedure is repeated with the new validation and training sets. We select the optimal hyperparameters based on which set gave the highest average classification accuracy across all fold combinations. We then use these set of hyperparameters and retrain the model on the entire training set (all three partitions) and then evaluate its final performance on the testing set. SVMs were trained using the of scikit-learn 1.3.0, along with numpy 1.25, and scipy 1.11 python libraries.

## Measurements of bacterial growth curves

For the final dataset (Supplementary Table 1, row 4), we used a library of 311 clinical *Enterobacteriaceae* isolates collected from the Duke University Hospital (supplied by Vance Fowler and Joshua Thaden) and North Carolina community hospitals (supplied by Deverick Anderson)[55]. These isolates have been described previously in multiple separate studies [Bioprojects PRJNA290784, PRJNA259658, and PRJNA551684][17,56,57]. They were collected as part of the ongoing Duke Infection Control Outreach Network (DICON) MDR Biorepository (since 2010) and the Duke Blood Stream Infection Biorepository (BSIB)

(since 2002). Both repositories contain prospectively collected Gram-negative organisms from hospitalized patients. The DICON bio-epository consists of isolates identified by microbiology laboratories at member hospitals as causing multidrug-resistant infections, while isolates in the BSIB were collected from patients with monomicrobial bacterial BSI at Duke University Hospital. These studies were approved by the Duke University Institutional Review Board. None of the clinical information related to these isolates, including how and where they were collected, was used in the present study.

The generation of the bulk of experimental data used in this work was described in Zhang et al.[17]. Frozen stocks were streaked on lyso-geny broth (LB) agar plates, and three separate colonies were selected to inoculate growth media. Overnight cultures were prepared in 1 mL of LB broth in 96-well deep-well microplates (VWR), which were shaken at 37 °C for 16 h at 1000 rpm. The OD600 (absorbance at 600 nm) for the overnight culture was taken on a plate reader (Tecan Spark). To ensure a consistent initial cell number, cultures were diluted to 1 $OD_{600}$ (assumed to be equivalent to $8 \times 10^8$ cells/mL) and further diluted 1:8 ($1 \times 10^8$ cells/mL). Cultures were then finally diluted 10-fold in 100 μL of fresh media in a 384-well deep-well plate (Thermo Scientific) using a MANTIS liquid handler for an initial cell density of $1 \times 10^6$ cells/well. LB media was used for all experiments, with three culture conditions: (1) no antibiotic treatment, (2) 50 μg/mL amoxicillin, and (3) 50 μg/mL amoxicillin + 25 μg/mL clavulanic acid.

Each of the three overnight cultures (three biological replicates, each from a distinct colony) were used to inoculate four wells for each condition (to generate four technical replicates), for a total of 12 replicates. The spatial position of all wells for each experiment was randomized across the plate to minimize plate effects. To minimize evaporation, the plate was loaded with the lid into the plate reader, which was equipped with a lid lifter, and the chamber temperature was maintained at 30 °C. $OD_{600}$ readings were taken every 10 min with periodic shaking (5 s orbital) for 24 h.

### Measurements of mixed-population growth curves

As with the clinical isolates, frozen stocks of the plasmid-free and plasmid-carrying laboratory strains used in mixed-population experiments were streaked on LB agar plates. Colonies were used to inoculate 2 mL LB in 16 mL culture tubes. 1 mM IPTG and 50 μg/mL kanamycin were added for overnights of plasmid-containing populations to maintain selection pressure and induce beta-lactamase production. Overnights were shaken at 37 °C for 16 h at 225 rpm. OD600 for the overnight cultures were measured; both cultures were corrected to an OD600 of 1. A total of 500 μL of plasmid-free and plasmid-carrying cells, respectively, were mixed. The resulting mixture was then diluted 1:16 for an assumed density of $5 \times 10^7$ cells/mL and 50 μL of 1 M IPTG was added to induce the beta-lactamase enzyme. Stock solutions of amoxicillin (Sigma) and a beta-lactamase inhibitor, either clavulanic acid (Sigma), tazo-bactam (Fisher Scientific), or sulbactam (Fisher Scientific), were prepared in DMSO (amoxicillin, tazobactam) or water (clavulanic acid, sulbactam) and were diluted into LB at concentrations 2.5 times the final concentrations. Using a MANTIS liquid handler, 40 μL of the appropriate antibiotic and inhibitor-containing solu-tions were dispensed into each well in a 384-well deep well plate (Thermo Scientific), followed by 20 μL of the diluted culture. Final initial cell density was $1 \times 10^6$ cells per 100 μL well, final IPTG con-centration was 1 mM, and final antibiotic and inhibitor concentra-tions formed a dose-response matrix of 0, 0.5, 1, 2, 4, 8, 16, 32, 64, and 128 μg/mL of the agents. Three replicate wells corresponded to each condition, and well positions were randomized across the plate. The plate was loaded with lid into the plate reader, and the chamber temperature was maintained at 30 °C. OD600 and GFP readings were taken every 10 min with periodic shaking (5 s orbital) for 24 h.

### Reporting summary
Further information on research design is available in the Nature Portfolio Reporting Summary linked to this article.

## Data availability
All data is available in the main text and the supplementary informa-tion are accessible at: https://github.com/yasab27/LSMGD.

## Code availability
All code is available in the main text and the supplementary informa-tion are accessible at: https://github.com/yasab27/LSMGD[58].

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

## Acknowledgements

We thank Daniel Cordray, Jerry Liu, Mirza Khalid Baig, Sumera Alam, Teng Wang, Kade Heckel, Abdullah Kuziez, Hunter Kemeny, Zachary Holmes, Anita Silver, and Zhengqing Zhou for lively discussion and comments on this work. This was work was partially supported by the Angier Buchanan Duke Memorial Scholarship (Y.B.), the National Institutes of Health (L.Y., R01AI125604, R01GM098642, and R01EB031869) and DARPA (L.Y. HR0011-23-2-0008).

## Author contributions

Y.B. and L.Y. conceived the research and designed the research framework. H.R.M. and L.Y. formulated the ODE model of the isolate antibiotic response. Y.B. performed all model simulation, neural network training, and machine learning analysis. H.R.M. and H.X. generated the experimental data. Y.B. and L.Y. wrote the paper with inputs from H.R.M. & H.X.

## Competing interests

The authors declare no competing interests.
