## [Peer Review File · Nature Communications]

REVIEWER COMMENTS

Reviewer #1 (Remarks to the Author):

The paper presents a new method for studying microbial communities using autoencoders to map their dynamics into a low-dimensional latent space. This approach enables the analysis of important biological properties, such as strain identity, antibiotic resistance, and predicting dynamical trajectories. The authors show that autoencoders can denoise growth curves without losing important biological information, which is crucial for accurate analysis and prediction. Additionally, the flexibility of the approach allows for adjusting the level of detail by choosing the appropriate embedding dimension. Overall, this new approach holds promise for advancing the study of microbial communities and could have important applications in the field.

Major:

* Can the authors show more phenotype applications than just AMR? The AMR example is still predicting from one isolate's growth curve not multiple so it does not seem to be that great of an advancement over reference 15. They need to show phenotype prediction from multi-member communities. Since the growth curves can be modeled — can they predict minimum inhibitory concentration (MIC) needed of the antibiotic rather than the classification of what type of antibiotic the strain has resistance to?

* There are some other methods for prediction of growth curves, <https://www.pnas.org/doi/10.1073/pnas.1902217116> - can the authors compare to this?

* Can they assess the robustness growth curve predictions to time-course perturbations (not just initial condition perturbations) with their dynamical systems setup?

* Can the authors compare the embedding distance of the growth dynamics for each sample -- that may lead to some interpretability on how different initial conditions or time dynamics can influence the embeddings?

* What is the interpretation of the embedding phenotypes? Would this model be able to still predict curves if there were horizontally transferred genes? Can this be used to define new phenotypes that are more accurate?

* Fig 2. C. What kind of reconstruction error is this? Caption says average MSE.. while figure has relative reconstruction error on the y-axis that varies from 0 to 1, so I think that it may be a percentage error. But at $E=2$, there isn't much visible error in the reconstruction, and one would think for a relative error of 100%, the reconstruction would not look at all like the true reconstruction — but it does. The error metrics are contradictory and not well explained and perhaps contradict how they relate to the reconstructions that we visualize. Please clarify.

* Predicting the growth dynamics from the initial conditions using the embeddings were all done with simulations. With datasets that exist for actual nutrient varying initial conditions, why don't the authors validate on real data? e.g. <https://doi.org/10.1371/journal.pcbi.1008817>

Minor:

* Line 42: ...using "them"... → ...using "the" ...

* Line 639: " is does" -> "does"

* line 197: "party" -> "Parity"

Reviewer #2 (Remarks to the Author):

Capturing microbial community dynamics in low dimensions is an exciting topic. Representing high-dimensional community dynamics in a lower dimension is valuable for understanding complex systems, such as microbial communities. The authors successfully employed an autoencoder to data mine microbial community dynamics. The computationally generated (growth curves) and experimental datasets were used to test how the autoencoder worked by reducing the data dimensions, e.g., 30 dimensions were sufficient to classify community members in the presence of

antibiotics. As the experimental datasets and the ML algorithm are previously reported and commonly known, respectively, I expect their combinations would lead to something new, e.g., novel findings on the microbial community. However, I could not find novel biological insights or new methodological know-how (point of view) in the present manuscript. If the authors could provide analytical results showing either the advantages and originality of their ML model/method or the novelty in (micro)biological findings, the readers would benefit from the study.

Major comments

1. The present ML method/model is better in compressibility but worse in interpretability. It's an achievement that the proposed method can represent higher dimensional dynamics in lower dimensions than the mechanical models. However, the mechanical models are good at reducing the dimensions with biologically interpretable parameters, e.g., discovering the biologically meaningful feature that governs the higher dimensional space. The present method is hard to interpret with biological meaning, which is assumed to be a critical disadvantage for using it in biological studies. As the method is supposed to be used in microbiology, biological interpretability should be considered.
2. The present study provides a first trial of using the autoencoder algorithm to analyze growth curves, which is an achievement. Nevertheless, it seems that no biological question/issue is considered in the present study, which might be why the data mining, e.g., classification and prediction, did not result in the novel finding.
3. The ability of the decoder to reconstruct community dynamics from low-dimensional features is the most significant impact of the present study, in my opinion. The authors should provide experimental or analytical evidence demonstrating the microbiological or computational value/benefit of the reproducibility; otherwise, reproducing the high-dimensional dynamics from the low dimensions is meaningless for researchers.

Minor comments

1. lines 43-45, add the references.
2. line 52, what does "human-engineered" mean?
3. lines 59-60, add the reference.
4. line 101, which in Figure 1A does "embedding vector" correspond to?
5. lines 118 - 119, it may be difficult to understand "The resultant embeddings exhibit a one-dimensional (1D) structure" from Figure 1B. Do you want to express that the data points ride on a $y=x$ for the two axes of Latent 1 and 2? If so, it would be better to describe it as such. As well as for line 126, "makes use of both dimensions" in Figure 1C.

6. lines 131-132, it is also difficult to understand "increasing the embedding dimension to $E = 10$ substantially improves the reconstruction quality" from Figure 1C, D. If you take the difference between the corresponding curves of "Original" and "Reconstruction", it can be more precise.
7. line 139 - 141, It is hard to imagine how to concentrate the growth curves. It would be better to show a diagram of the concatenate growth curves as in Supplementary Figure 1B.
8. line 162, what does the "full data sets" refer to?
9. line 166, SVM, spell it out when using it for the first time.
10. line 171, What is "the highest confidence prediction"? Can you provide numerical data?
11. line 188, what does the "original raw data" refer to?
12. line 196, It is difficult to understand "representing ~ 10 -fold data compression". Does it mean that the original dimension of $D=98$ has been reduced to $E=10$, and so the dimension has been reduced to $1/10$?
13. line 210, what does "We simulated a two-member microbial community" mean for a microbial community? Is it their growth dynamics?
14. lines 211 - 212, what are the parameters of Species1 and Species2 on the vertical and horizontal axes in Figure 4B?
15. lines 282 - 283, how does Figure 6B lead to the statement?

Responses to reviewers' comments

Unless noted otherwise, all figures mentioned below follow the indexing in the revised manuscript.

Review 1:

Overview

The paper presents a new method for studying microbial communities using autoencoders to map their dynamics into a low-dimensional latent space. This approach enables the analysis of important biological properties, such as strain identity, antibiotic resistance, and predicting dynamical trajectories. The authors show that autoencoders can denoise growth curves without losing important biological information, which is crucial for accurate analysis and prediction. Additionally, the flexibility of the approach allows for adjusting the level of detail by choosing the appropriate embedding dimension. Overall, this new approach holds promise for advancing the study of microbial communities and could have important applications in the field.

We thank the reviewer for recognizing novelty of our work and its implication for advancing the study of microbial communities. We also thank the reviewer for providing detailed comments on technical aspects of our analysis. We have conducted extensive analysis to address these points.

Major Comments:

1. Can the authors show more phenotype applications than just AMR? The AMR example is still predicting from one isolate's growth curve not multiple so it does not seem to be that great of an advancement over reference 15. They need to show phenotype prediction from multi-member communities. Since the growth curves can be modeled — can they predict minimum inhibitory concentration (MIC) needed of the antibiotic rather than the classification of what type of antibiotic the strain has resistance to?

We thank the reviewer for these insightful comments and suggestions. The key to using autoencoder (including the basic autoencoder (Figure 1) and the variational autoencoder (Supplementary Figure 4) is to achieve unbiased compression of the raw data into a low-dimensional latent space. This latent space can be used in a variety of manners, including classification of traits and prediction of quantitative traits.

In our original manuscript, we have demonstrated the use of the latent space for classification, which includes both strain ID prediction and resistance prediction. In both cases, the outputs are categorical. The major advance over the previous study (Zhang et al, 2020)¹ is the improved performance by using the latent space, despite the drastic reduction of dimensionality in data.

We have conducted two major sets of analysis to incorporate the reviewer's suggestions on making quantitative inference and predictions.

1. ML-assisted parameter estimation from experimental data

Here, we demonstrate the use of the VAE combined with kinetic modeling to estimate model parameters from experimental data. In doing so, we have developed a new pipeline as below:

1. Formulate a kinetic model that has the capacity to describe experimental data.
2. Generate an ensemble of simulations by randomly sampling model parameters.
3. Train a VAE with a small latent dimension using the simulated curves.
4. Map the latent space of the VAE to the model parameters by training a separate neural network (a multi-layer perceptron, or MLP).
5. Use the combined VAE-MLP trained with simulation data to estimate kinetic parameters using experimental data generated from 311 clinical isolates.

As shown in our new results (Figure 4), the pipeline enables highly effective estimation of kinetic parameters. It also provides concrete evidence for the presence of sloppy parameters, which we discussed in the original submission. In particular, the pipeline enabled accurate estimate (judged by the simulated data, where we have ground truth) some but not all parameters. Even though some parameters

were not well estimated, the estimated parameters were able to reproduce the system dynamics with high fidelity. We have reproduced the associated figure and caption below for convenience:

Figure 4 | VAE latent spaces can map to mechanistic growth parameters

(A) Mapping from growth curves to parameters. The encoder from a trained VAE is used to map growth curves to a low dimensional latent space. A multilayer perceptron is then trained to estimate model parameters from the latent space. The VAE and MLP models are trained using simulated data with a wide range of growth parameters selected at random from a distribution over a biologically plausible range. The individual training examples correspond to three growth curves generated using an ODE model incorporating the effects of beta-lactam antibiotic and Bla-inhibitor as in Figure 2 (Methods). Each curve corresponds to a different combination of initial concentrations of the antibiotic and *Bla* inhibitor.

(B) The neural network enables accurate estimated of some but not all parameters. We show the comparison of five estimated parameters vs the ground for training (blue) and test (orange) sets of growth curves. For the first four parameters can be estimated with higher accuracy. The fifth parameter, κ_b , is poorly estimated, suggesting it is a ‘sloppy’ parameter.

(C) Estimated parameters generate accurate predictions of the growth dynamics. We apply our latent-space mapping procedure to experimental growth curves used in Figure 2 (Supplementary Table 1). Despite being only trained on simulated growth curve, NN-estimated parameters from the experimental growth curves can enable the mechanistic model to predict growth curves with high fidelity.

2. Predict strain and experimental configurations of a two-member microbial community using experimental data.

In addition, we have conducted extensive additional experiments to generate temporal dynamics of two-member community using two readouts (total biomass measured in terms of OD and fluorescence associated with one member), in response to different drug type- and dose-combinations. In total, we generated 9200 time courses for this new analysis. This set of experiments was critical as we did not find openly available data from the literature that have sufficient quantity or temporal resolution to conduct our ML analysis. We will make this experimental data set openly available for the research community.

Using these data, we demonstrated the prediction of different combinations of strains, drug types, and drug doses using the time courses, through the VAE-deduced latent space. Briefly, our ML pipeline includes the following steps:

1. Train a VAE model using the experimentally measured time courses (OD and GFP).
2. Build a classifier using the VAE-deduced latent variable to predict combinations of strains, plasmids, and drug types (Supplementary Figure 7).

These new analyses further demonstrate the versatile uses of the latent space deduced by a properly trained VAE model.

2. There are some other methods for prediction of growth curves, <https://www.pnas.org/doi/10.1073/pnas.1902217116> - can the authors compare to this?

We thank the reviewer for highlighting this work, which we have cited in the revised manuscript. Briefly, this paper focuses on predicting microbial growth in a mixed two-member community by estimating the parameters from individual growth dynamics of the two species in monocultures. The approach requires sufficient prior knowledge of the system to formulate the underlying kinetic model properly. In addition, the approach requires that the parameters estimated from clonal populations are sufficiently maintained in the mixture – i.e., the interaction between the two populations is not too strong.

Our approach (and ML-based approach in general) complements such kinetic-model-guided approach. Specifically, our work aims to deduce the essential dimension underlying apparently complex community dynamics, by using an autoencoder with varying sizes of the latent space. Once this latent size is defined, which is constrained by desired reconstruction fidelity (Figure 2), it can be used in various applications, depending on the context. For instance, it can be used to predict the system dynamics from different initial conditions (Figure 5) or different experimental conditions (Figure 7).

The deduction of the latent space is entirely data-driven and does not require a mechanistic basis. However, when an underlying mechanistic model is available, the latent space can be mapped to mechanistic model parameters and vice versa. As illustrated above (Figure 4), this mapping can allow us to estimate kinetic parameters from experimental data.

We have revised the main text to clarify these points.

3. Can they assess the robustness growth curve predictions to time-course perturbations (not just initial condition perturbations) with their dynamical systems setup?

We thank the reviewers for their suggestion. Accordingly, we introduce new analysis and applying our method to predict the dynamics of a simulated microbial community with random antibiotic perturbations of various amplitudes and time values. We show that our MLP-VAE approach can be adapted to predict the dynamics of these systems from the time and concentration of antibiotic perturbation with high accuracy. We include a supplementary figure and discussion detailing this analysis in our revised manuscript, reproduced below:

Supplementary Fig. 6 | VAE enables prediction of growth dynamics resulting from temporal perturbations.

(A) Mapping temporal perturbations to dynamics. We generated a dataset of 5000 simulated 3-member community dynamics using a gLV model. Each sample consists of growth curves (each of 50 data points) of the 3 members. The model parameters and initial compositions were fixed for all simulations. The community dynamics were perturbed by a fixed duration pulse of antibiotic added at variable doses and starting times. To predict community dynamics, as with the initial-condition-to-trajectory model, we first train a VAE on the community dynamics to generate a low dimensional latent space to represent these dynamics. We then train an MLP that maps the parameters of the antibiotic perturbation (antibiotic dosage and time of perturbation) to a corresponding point in the latent space, which is in turn decoded to get a prediction of the community dynamics.

(B) VAE-MLP enables high accuracy prediction of growth dynamics from temporal perturbations. Even with a low latent dimension of $E = 5$, the MLP-VAE mapping predicted community dynamics with a high accuracy. As the embedding dimension is increased, the

accuracy improves as evidenced by the shrinking scatter about the line $y = x$. Here we show the predictions in the test dataset.

4. Can the authors compare the embedding distance of the growth dynamics for each sample -- that may lead to some interpretability on how different initial conditions or time dynamics can influence the embeddings?

We thank the reviewer for this insightful suggestion. In our original manuscript, Figure 4 (now moved to Supp Figure 4) analyzes the structure and distance within the latent space for predicting the dynamics of a two-member simulated microbial community from various initial conditions. In this system, the VAE embeds the community growth dynamics into an isotropic Gaussian distribution. We show that points which are located an equal distance from the center of Gaussian are interpolated by our decoder to points in phase space of comparable arc-length distance to the fixed-point attractor of the phase space. This suggests the embedding dimension learns a nonlinear transformation of the phase space of the system. This embedding approximately represents polar coordinates centered around the fixed point of our system.

5. What is the interpretation of the embedding phenotypes? Would this model be able to still predict curves if there were horizontally transferred genes? Can this be used to define new phenotypes that are more accurate?

We thank the reviewer for these insightful questions. It is difficult to interpret the VAE embeddings in isolation, in part due to the way it is deduced. The values of the latent variables are constrained by several factors associated with neural network training, including: the neural network architecture and parameterization, the size of the latent space, as well as the hyperparameters during training (e.g. batch size and learn rate). Two different VAEs with the same size of the latent space can achieve similar reconstruction fidelity using different sets of latent variables.

However, once they are deduced, the latent embeddings can be mapped to variables and conditions that have concrete interpretations. For instance, we have demonstrated that the latent embeddings can be mapped to the initial conditions (Figure 5), kinetic model parameters (Figure 4), and experimental configurations (Figure 6, 7, Supplementary Figures 6, 7).

On the question of horizontal gene transfer, we include additional analysis applying our VAE-MLP method to predict the growth dynamics of simulated microbial communities which exchange plasmids. Specifically, we consider the growth dynamic of a two-member microbial communities transferring two plasmids that confer varying fitness effects (burden or benefit). In our supplementary figure and description, we show that our model can predict the growth dynamics of the microbial populations from the initial distributions of plasmids with high accuracy:

a

$$\frac{ds_i}{dt} = \alpha_i \mu_i^c s_i - D s_i$$

$$\frac{dp_{ij}}{dt} = \beta_{ij} \mu_{ij}^c p_{ij} + (s_i - p_{ij}) \sum_{k=1}^m \eta_{jki} p_{kj} - (\kappa_{ij} + D) p_{ij}$$

Supplementary Fig. 5 | VAE enables dynamics prediction for communities with HGT.

(A) Horizontal gene transfer model. We simulate the dynamics of a two-member microbial community trading two plasmids using a plasmid centric framework. Briefly, we separately simulate the population dynamics of the individual species i and within each species, the population carrying a given plasmid j . The resulting system is thus captured by 2 ODEs describing the species dynamics and 4 ODEs corresponding to the dynamics of plasmids being carried within each population (Methods). Using this formulation, we generate a dataset of 5000 community growth curves by varying the initial concentration of each plasmid while holding all other system parameters and initial conditions fixed.

(B) Two step initial condition-to-dynamics mapping. As with the other prediction problems, our procedure consists of first training a variational autoencoder to learn a low dimensional embedding of our system dynamics. A multilayer perceptron is then used to predict the species population dynamics from initial distribution of plasmids within the species. Note, though our ODE model resolves the dynamics of the both the species and plasmids, we train our variational autoencoder using only species dynamics, so the model has no explicit information on the dynamics of horizontal gene transfer except through their implicit impacts on population dynamics.

(C) VAE-MLP enables high accuracy prediction of community dynamics. Despite not having any information on the plasmid dynamics, we see the VAE-MLP can predict the community dynamics of both species from initial plasmid concentration to high accuracy even at very low dimension of $E = 2$. As the embedding dimension improves, the quality of prediction increases, though slightly.

6. Fig 2. C. What kind of reconstruction error is this? Caption says average MSE.. while figure has relative reconstruction error on the y-axis that varies from 0 to 1, so I think that it may be a percentage error. But at $E=2$, there isn't much visible error in the reconstruction, and one would think for a relative error of 100%, the reconstruction would not look at all like the true reconstruction — but it does. The error metrics

are contradictory and not well explained and perhaps contradict how they relate to the reconstructions that we visualize. Please clarify.

We thank the reviewer for this comment and have clarified the figure and caption in question. We have changed the y-axis to mean square error, averaged over all training examples. This value shrinks with increasing embedding dimension and highlights how as we increase the dimension of the embedding, the better the reconstruction quality.

7. Predicting the growth dynamics from the initial conditions using the embeddings were all done with simulations. With datasets that exist for actual nutrient varying initial conditions, why don't the authors validate on real data? e.g. <https://doi.org/10.1371/journal.pcbi.1008817>

We thank the reviewer for this suggestion. The suggested reference is highly relevant, and we have cited in the revised manuscript. However, the amount of data in the study was insufficient to train a VAE for quantitative predictions.

To incorporate the reviewer's suggestion, we have conducted extensive new experimental analysis and generated a dataset of 9200 time courses corresponding to two member microbial community of varying component strains, plasmids, antibiotic, and antibiotic inhibitor concentrations. We show that we can with high accuracy predict the dynamics of the two-member population across conditions from system these system parameters. We have updated our manuscript and included a new figure to showcase these results.

8. Minor comments:

* Line 42: ...using "them"... \ ...using "the" ...

* Line 639: " is does" -> "does"

* line 197: "party" -> "Parity"

We thank the reviewer for this comment and have corrected these typos within the manuscript.

Review 2 Overview

Capturing microbial community dynamics in low dimensions is an exciting topic. Representing high-dimensional community dynamics in a lower dimension is valuable for understanding complex systems, such as microbial communities. The authors successfully employed an autoencoder to data mine microbial community dynamics. The computationally generated (growth curves) and experimental datasets were used to test how the autoencoder worked by reducing the data dimensions, e.g., 30 dimensions were sufficient to classify community members in the presence of antibiotics. As the experimental datasets and the ML algorithm are previously reported and commonly known, respectively, I expect their combinations would lead to something new, e.g., novel findings on the microbial community. However, I could not find novel biological insights or new methodological know-how (point of view) in the present manuscript. If the authors could provide analytical results showing either the advantages and originality of their ML model/method or the novelty in (micro)biological findings, the readers would benefit from the study.

We thank the reviewer for recognizing the value of capturing microbial community dynamics in low dimensions, which is the focus of our work. As noted by the review, the ability to do so is valuable for understanding complex systems like microbial communities. We are also glad that reviewer find our demonstration successful (on the technical front).

We regret the lack of clarity in our original manuscript on the conceptual significance of our analysis. By demonstrating the drastic dimension-reduction of dynamics of clonal or community dynamics (computational and experimental), our work generated the following insights in terms of the overall properties of microbial communities as dynamical systems.

Microbial community growth dynamics can be represented using far fewer degrees of freedom than what is needed for a mechanistic model. As evidenced by the ability to both compress and reconstruct both experimental and simulated microbial growth curves from low dimensional embedded spaces with high fidelity, microbial growth curves possess an internal "latent structure" which can be captured by neural networks to distill a reduced-dimension summary of a growth curve. The ability to capture temporal dynamics with lower parameter representations than the number of parameters required for ODE models suggests a substantial degree of redundancy or "sloppy" parameters in these mechanistic models.² **This property has implication for modeling, measuring, and controlling dynamics of microbial communities.**

1. The present ML method/model is better in compressibility but worse in interpretability. It's an achievement that the proposed method can represent higher dimensional dynamics in lower dimensions than the mechanical models. However, the mechanical models are good at reducing the dimensions with biologically interpretable parameters, e.g., discovering the biologically meaningful feature that governs the higher dimensional space. The present method is hard to interpret with biological meaning, which is assumed to be a critical disadvantage for using it in biological studies. As the method is supposed to be used in microbiology, biological interpretability should be considered.

We thank the reviewer for the insightful comment. The reviewer is correct that the autoencoder achieves data compression without directly enabling direct interpretability. Indeed, it is difficult to interpret the VAE embeddings in isolation, in part due to the way they are deduced. The values of the latent variables are constrained by several factors associated with neural network training, including: the neural network architecture and parameterization, the size of the latent space, as well as the hyperparameters during training (e.g. batch size and learn rate). Two different VAEs with the same size of the latent space can achieve similar reconstruction fidelity using very different sets of latent variables.

However, once they are deduced, the latent embeddings can be mapped to variables and conditions that have concrete interpretations. For instance, we have demonstrated that the latent embeddings can be mapped to the initial conditions (Figure 5), kinetic model parameters (Figure 4), and experimental configurations (Figure 6, 7, Supplementary Figures 6, 7). On the technical front, this mapping can be

more effective than direct use of the original, uncompressed data, as evidenced by the improved performance in strain ID and resistance prediction using the latent embeddings (Figures 3, Supplementary Figure 7). On the conceptual front, the mapping provides an indirect manner to interpret the latent variables, through the parameters or system configurations they are mapped to.

2. The present study provides a first trial of using the autoencoder algorithm to analyze growth curves, which is an achievement. Nevertheless, it seems that no biological question/issue is considered in the present study, which might be why the data mining, e.g., classification and prediction, did not result in the novel finding.

We thank the reviewer for recognizing our technical achievement. We regret not having articulated the conceptual contribution sufficiently well in the original submission. The key question we aim to address is the effective dimension of a microbial community as a dynamical system. Using autoencoders, we find that the apparently complex temporal dynamics of clonal or community dynamics can be captured by only a few variables (latent variables). For a multi-member microbial community, the number of latent variables needed to describe the community dynamics can be fewer than the number of parameters needed for a standard gLV model. This is the major biological insight emerging from our analysis.

We have revised our manuscript to clarify this point.

3. The ability of the decoder to reconstruct community dynamics from low-dimensional features is the most significant impact of the present study, in my opinion. The authors should provide experimental or analytical evidence demonstrating the microbiological or computational value/benefit of the reproducibility; otherwise, reproducing the high-dimensional dynamics from the low dimensions is meaningless for researchers.

Indeed, the central conceptual contribution is that community dynamics can be constructed from highly concise, low-dimensional features (latent embeddings). Once deduced, these latent embeddings serve a gateway to versatile downstream analyses.

The ability to do so has several practical benefits and values in computational and analysis, which we highlight below:

1. In our original manuscript, we demonstrated that these latent embeddings are more effective in predicting strain ID and drug resistance than using the full growth curves (Figure 3).
2. The deduction of the latent embeddings is data driven and does not require an underlying mechanistic model. However, if such a model is available, the combination of the kinetic modeling and machine learning can enable highly efficient parameter estimation using experimental data. We demonstrate this point by establishing a new pipeline for parameter estimation:
 - a. Formulate a kinetic model that has the capacity to describe experimental data.
 - b. Generate an ensemble of simulations by randomly sampling model parameters.
 - c. Train a VAE with a small latent dimension using the simulated curves.
 - d. Map the latent space of the VAE to the model parameters by training a separate neural network (a multi-layer perceptron, or MLP).
 - e. Use the combined VAE-MLP trained with simulation data to estimate kinetic parameters using experimental data generated from 311 clinical isolates.

As shown in our new results (Figure 4), the pipeline enables highly effective estimation of kinetic parameters. It also provides concrete evidence for the presence of sloppy parameters, which we discussed in the original submission. In particular, the pipeline enabled accurate estimate (judged by the simulated data, where we have ground truth) some but not all parameters. Even though some parameters were not well estimated, the estimated parameters were able to reproduce the system dynamics with high fidelity.

3. Also, by applying the VAE to simulated community dynamics, we demonstrate that the effective number of variables to reconstruct the community dynamics can be fewer than the number of parameters needed to formulate a standard gLV model. The number of latent variables necessary for reliable reconstruction scales approximately linear with the number of community members (n), whereas the number of gLV model parameters scales with n^2 (Figure 6). This result indicates a substantial sloppiness in the gLV model parameters. The insight is non-intuitive and difficult to establish without using ML.

Our result suggests that ML can be used as a generalizable approach to probe the sloppiness of dynamical models, regardless of how the models are formulated.

Minor comments

1. lines 43-45, add the references.

We thank the reviewer for this comment and have added appropriate references accordingly.

2. line 52, what does "human-engineered" mean?

By human-engineered, we refer to the use of hand-crafted features for summarizing the growth curves, such as growth integral or growth rate, that are designed and chosen by the researcher. This is as differentiated from using an autoencoder to summarize the data, which generates low dimensional features to summarize growth curves directly from data.

In light of the reviewer's comment, we realize that this term can be confusing and we rephrased the term to "pre-selected".

3. lines 59-60, add the reference.

We thank the reviewer for this comment and have added appropriate references accordingly.

4. line 101, which in Figure 1A does "embedding vector" correspond to?

In figure 1A, the embedding vector corresponds to the latent vector z . We have updated the figure caption to improve clarity on this point.

5. lines 118 - 119, it may be difficult to understand "The resultant embeddings exhibit a one-dimensional (1D) structure" from Figure 1B. Do you want to express that the data points ride on a $y=x$ for the two axes of Latent 1 and 2? If so, it would be better to describe it as such. As well as for line 126, "makes use of both dimensions" in Figure 1C.

6. lines 131-132, it is also difficult to understand "increasing the embedding dimension to $E = 10$ substantially improves the reconstruction quality" from Figure 1C, D. If you take the difference between the corresponding curves of "Original" and "Reconstruction", it can be more precise.

We take the response to both comments 5 and 6 together. We regret the confusion. We have reworked Figure 1, which demonstrates the basic principle of encoding and decoding. To simplify the analysis, we have opted to use full growth curves instead of their derivatives. To point 5, we have updated the text of our results section to clarify what is meant by one-dimensional structure and leveraging two-dimensional structure. To address point 6, we have changed Figure 1C and 1D to show the embedding and reconstruction of the same set of initial growth curves. This approach highlights the difference in quality of reconstruction between $E = 2$ and $E = 10$ case. We have also updated the text of our analysis to compare the mean absolute error between the $E = 2$ and $E = 10$ to quantitatively measure the difference in reconstruction between the two sets of reconstructions.

7. line 139 - 141, It is hard to imagine how to concatenate the growth curves. It would be better to show a diagram of the concatenate growth curves as in Supplementary Figure 1B.

We thank the reviewer for the suggestion. We have updated Figure 2 accordingly with a diagram showing our feature engineering and concatenation procedure.

8. line 162, what does the "full data sets" refer to?

We thank the reviewer for this comment. By "full data sets" we meant the usage of an entire time course for analysis instead of a neural network embedding. We have clarified the language in this section to indicate this.

9. line 166, SVM, spell it out when using it for the first time.

We thank the reviewer for this comment and have clarified the language accordingly.

10. line 171, What is "the highest confidence prediction"? Can you provide numerical data?

By "highest confidence prediction", we are describing the procedure used to achieve multiclass classification, in this case classifying between multiple different strains of bacteria. In a multiclass classification strategy, separate binary machine learning classifiers are trained for to differentiate each class (here the 311 strains). To classify a growth curve into one of these strains, each trained binary classifier computes a confidence score that a particular sample belongs into its class or outside of it. For an SVM, this prediction is based on distance from a hyperplane separating the two groups, with a larger distance from the hyperplane indicating higher likelihood that a given sample has been correctly classified. To select a final classification label for a growth curve, we choose the label from the SVM which gives the highest confidence score. The confidence scored is based on the difference from the hyperplane.

We have revised our manuscript text to better clarify this point.

11. line 188, what does the "original raw data" refer to?

We thank the reviewer for this comment. By "original raw data" we meant the usage of an entire time course for analysis instead of a neural network embedding. We have clarified the language in this section to indicate this.

12. line 196, It is difficult to understand "representing ~10-fold data compression". Does it mean that the original dimension of $D=98$ has been reduced to $E=10$, and so the dimension has been reduced to $1/10$?

We thank the reviewer for this comment. Yes, this is what we meant by 10-fold compression. We have clarified the text accordingly.

13. line 210, what does "We simulated a two-member microbial community" mean for a microbial community? Is it their growth dynamics?

We thank the reviewer for this comment. Yes, we meant growth dynamics. We have clarified the language of the text accordingly.

14. lines 211 - 212, what are the parameters of Species1 and Species2 on the vertical and horizontal axes in Figure 4B?

The growth rate parameters μ_i used to generate the original Figure 4B (now **Supplementary Figure 4**) were sampled i.i.d. at random from a uniform distribution $[0,1]$ and the interaction parameters from γ_{ij} from a $N(0,1)$. The background stress parameters were fixed to $\sigma = 0.05$. We did not record the exact parameter values used to generate the simulations, but we found that our results in Supplementary Figure 4B were consistent for any choice of μ_i and γ_{ij} .

15. lines 282 - 283, how does Figure 6B lead to the statement?

We regret the confusion. Figure 6B shows the comparison between predicted and ground-truth values. Their alignment improved from $E = 3$ to $E = 6$, as indicated by the reduced scattering of points around the line $y = x$, as well as the increased R^2 values. This improved performance is also evident in the mean square errors, shown for the 3-member community. It shrinks rapidly with increasing E before plateauing at $E = 4$.

We have revised the main text to better explain the trend.

- 1 Zhang, C. *et al.* Temporal encoding of bacterial identity and traits in growth dynamics. *Proc Natl Acad Sci U S A* **117**, 20202-20210 (2020). <https://doi.org:10.1073/pnas.2008807117>
- 2 Gutenkunst, R. N. *et al.* Universally sloppy parameter sensitivities in systems biology models. *PLoS Comput Biol* **3**, 1871-1878 (2007). <https://doi.org:10.1371/journal.pcbi.0030189>

REVIEWERS' COMMENTS

Reviewer #1 (Remarks to the Author):

concerns addressed

Reviewer #2 (Remarks to the Author):

The authors carefully revised their manuscript to avoid over-interpretation or misleading, which addressed the questions I raised to some extent (Figs. 4 and 7). The main issue that remains to be considered is the biological usefulness of the method demonstrated by the actual growth data.

Duke University

DURHAM
NORTH CAROLINA
27708-0281

DEPARTMENT OF BIOMEDICAL ENGINEERING
PRATT SCHOOL OF ENGINEERING
OFFICE 2355, CIEMAS
BOX 3382
101 SCIENCE DRIVE, DURHAM, NC 27708

TELEPHONE (919) 660-8408
FAX (919) 668-0795
EMAIL: you@duke.edu

Responses to REVIEWERS' COMMENTS

Reviewer #1 (Remarks to the Author):

concerns addressed

Reviewer #2 (Remarks to the Author):

The authors carefully revised their manuscript to avoid over-interpretation or misleading, which addressed the questions I raised to some extent (Figs. 4 and 7). The main issue that remains to be considered is the biological usefulness of the method demonstrated by the actual growth data.

Response:

We are glad that our revision and clarifications have addressed most of the reviewers' concerns. While these lingering points were addressed in our previous response letter, we agree with the reviewer that they need to be more clearly explained in the manuscript. As such, we have added a paragraph in the Discussion section to explain the limitations and uses of autoencoders to deduce concise representations of microbial community dynamics. The additional paragraph will avoid over-interpretation or misleading interpretation of the latent space, despite its versatile utilities.